# Comprehensive multi-metric analysis of user experience and performance in adaptive and non-adaptive lower-limb exoskeletons

Krongkaew Supapitanon[1], Tanyaporn Patathong[1], Chaicharn Akkawutvanich[2], Arthicha Srisuchinnawong[2], Worachit Ketrungsri[2], Poramate Manoonpong[2], Patarawan Woratanarat[1], Chanika Angsanuntsukh[1] *

1 Department of Orthopedics, Faculty of Medicine Ramathibodi Hospital, Mahidol University, Ratchathewi, Bangkok, Thailand, 2 School of Information Science & Technology, Vidyasirimedhi Institute of Science & Technology (VISTEC), Rayong, Thailand

* achanika@gmail.com

## Abstract

Among control methods for robotic exoskeletons, biologically inspired control based on central pattern generators (CPGs) offer a promising approach to generate natural and robust walking patterns. Compared to other approaches, like model-based and machine learning-based control, the biologically inspired control provides robustness to perturbations, requires less computational power, and does not need system models or large learning datasets. While it has shown effectiveness, a comprehensive evaluation of its user experience is lacking. Thus, this study addressed this gap by investigating the performance of a state-of-the-art adaptive CPG-based exoskeleton control system (intelligent mode) under a multi-metric analysis (involving three-dimensional gait analysis, muscle activity, oxygen consumption, user comfort, and exoskeleton performance scores) and comparing it to a standard commercial exoskeleton control system (default mode). A cross-over design with randomized allocation in Thai healthy and independently walking adults ensured participants experienced both modes. All participants were assigned into two groups to receive an alternate sequence of walking with the intelligent mode or the default mode of the lower-limb exoskeleton Exo-H3 at high and normal speed. From eight participants, the intelligent mode-driven exoskeleton (adaptive exoskeleton) showed a significantly lower velocity, stride, and step lengths than the default mode-driven exoskeleton (non-adaptive exoskeleton). This setup significantly increased anterior pelvic tilt during mid-swing at normal speed (3.69 ± 1.77 degrees, p = 0.001) and high speed (2.52 ± 1.69 degrees, p = 0.004), hip flexion during stance phase with ankle dorsiflexion, and used less oxygen consumption at high speed (-2.03 ± 2.07 ml/kg/min) when compared to the default one. No significant differences of muscle activity, user comfort and exoskeleton performance scores between the two modes. Further exoskeletal modification in terms of hardware and control is still needed to improve the temporal spatial, kinematics, user comfort, and exoskeleton performance.

**Data Availability Statement:** All relevant data are within the manuscript and its Supporting information files.

**Funding:** This work was supported by the Vidyasirimedhi Institute of Science & Technology (VISTEC) under the EXOVIS project (Grant No. I20POM-INT010).

**Competing interests:** The authors have declared that no competing interests exist.

## 1. Introduction

Medical assistive devices have been variously evolved to improve human moving abilities. Regarded as empowering technology, exoskeletons or robot suits have complemented walking and climbing support for many decades [1]. These appliances also extend, substitute, and enhance function, as well as play an important role in the treatment and rehabilitation of movement disorders [2]. Several studies found the lower-limb exoskeleton training was generally safe and feasible for the patient with gait problem such as spinal cord injury, stroke, cerebral palsy [3–8].

The effectiveness of exoskeleton training depends on individual factors such as the smoothness of its movement and coordination. Thus, many studies [9] were focusing not only on hardware design, but also a development of efficient control algorithm. Generally, the control strategies can be classified into high-, mid-, or low-level control layers [10]. The high-level layer provides operation mode to the crucial mid-level layer where gait detection and synchronization occur. A basic control method depends on predefined joint trajectories and initiates different walking pattern by a user [11]. To increase adaptability in different conditions, some controls implemented gait planning based on a kinematic model [12] or a simplified nonlinear dynamic model as inverted pendulum [13]. The generated profiles were then controlled by different real-time trajectory tracking methods such as proportional-integral-derivative (PID) control integrated in the low-level motor driver or linear quadratic regulator (LQR) control. Advanced control strategies were introduced such as PID nonlinear controller with Linear Matrix Inequalities [14], backstepping control technique [15], and sliding mode control [16], to guarantee stability and robustness. However, those model-based methods required solving complex equations during the offline process. Apart from joint profile, there was torque related control scheme [17]. Nevertheless, it required a torque sensor for feedback to gain more accurate control over several situations.

Strictly to joint angle sensor, adaptive-oscillator-based or central pattern generator (CPG)-based control, which was considered as a biologically-inspired model-free approach, was introduced to handle rhythmic movement as walking. Its intrinsic property as limit cycle provided a stable joint trajectory and disturbance rejection by nature. Adaptive frequency oscillator (AFO) [18] was proposed to entrain frequency from feedback. It required fewer trails (~ six gait cycles) to adapt for individual user. Another recent adaptive trajectory/shape generation framework was dynamic movement primitive and hierarchical interactive learning (DMP-HIL) [19]. It tried to solve a synchronization problem by modeling as leader-follower agents. The method required approximately 15 gait cycles to converge. More techniques had been added to the aforementioned framework and turned it into coupled cooperative primitive (CCP) framework [20], which coupled the interaction model in both the velocity and acceleration levels. It has also been proven to outperform the conventional DMP-HIP in term of interaction torque reduction. However, the method could still not adapt to different frequencies.

To achieve online frequency and shape adaptations in response to a joint angle tracking error, our study [21] modified the previous CCP framework with CPG module called SO(2) oscillator [22] whose inputs were associated with gradients of the joint angle tracking error. This novel control was called an adaptive modular neural control (AMNC) with less adaptation time (within a gait cycle). Although, the proposed method for online gait synchronization effectively optimized the tracking error by 80% and interaction torque by 30% [21], a comprehensive evaluation of its effectiveness in term of spatiotemporal and biological signals on practical situation in real clinical environment has not been demonstrated before. This study addressed this gap by testing a state-of-the-art adaptive biologically-inspired control system based on CPGs (intelligent mode) [21], and comparing its performance to a standard

commercial system (default mode) across multiple metrics, including three-dimensional gait analysis, electromyography, oxygen consumption, user comfort, and exoskeleton performance scores. Expected benefits of this adaptive version would ameliorate natural walking and essential lower limb motions.

## 2. Methods

### 2.1 Study design

A 2x2 cross-over design of the development of exoskeleton's control system in assistive mode compared between the default mode and the intelligent mode in normal Thai people was conducted between September 2021 and February 2024 at Faculty of Medicine of Ramathibodi Hospital. Each participant provided informed written consent before being included in the study. The individual pictured in S3 File has provided written informed consent (as outlined in PLOS consent form) to publish their image alongside the manuscript. This study protocol was approved by Human Research Ethics Committee of Faculty of Medicine of Ramathibodi Hospital (COA. MURA2021/260, see S1 Text), and supporting CONSORT checklist are available as supporting information (see S1 Checklist). The trial protocol was retrospectively registered with clinicaltrials.gov identifier: NCT06513390.

### 2.2 Participants

Participants were recruited in July 2022. Healthy at Thai adults were eligible if their age between 18 and 60 years old, body mass index between 18.5 and 24.9 kg/m$^2$, able to walk independently, and willing to participate in the study. Participants were excluded for the following reasons: (1) a history of surgery in the back, hip, knee, or ankle area; (2) previous injury or pain in the hip, knee, or ankle area that would affect walking patterns within the past 6 months; (3) a history of musculoskeletal or neuromuscular disease such as multiple sclerosis, myasthenia gravis; (4) a balance disorder; and (5) unable to continue or withdrawal from the study.

### 2.3 Randomization

The sequence was randomized using a simple randomization technique, conducted by a statistician not involved in the study. Eligible participants were randomly assigned into two groups: group 1 began with walking in the default mode (intervention A) and then switched to the intelligent mode (intervention B), while group 2 started with the intelligent mode followed by the default mode. Allocation was concealed in the opaque envelops prior to the exoskeletal application, and participant was blinded to the sequence of the two walking conditions.

### 2.4 Instrument

The exoskeleton used in this experiment was the 11-kg Exo-H3 (Technaid SL, Arganda del Rey, Spain), see S1 File. This wearable device contains six actuated joints at the hip, the knee, and the ankle on both legs accompanied with direct current (DC) motors and harmonic gears. Range of motion were set at 30˚ backward and 105˚ forward for the hips, 105˚ backward to 5˚ forward for the knees, and 30˚ upward and downward for the ankles. The maximum operating torque for each joint is approximately 40 Nm. The Exo-H3 provides assistive capabilities for users weighing up to 100 kg by incorporated joints and interaction torque sensors quantifying the user-exoskeleton direct force. The exoskeleton hip orthosis supports the wearer's trunk, while adjustable coated aluminum alloy bars restrict the thigh, shank, and major leg structures. To maintain balance movement in the frontal plane, the wearer utilized forearm crutches.

## 2.5 Intervention

**Default mode.**    A basic commercial control system offering a conventional assistive function is employed for gait generation. Bilateral hip-knee-ankle joint position patterns (θ) represent average European subjects' profiles. A mid-level control algorithm samples the joint profile and transmits each joint angle data (θ) at an appropriate time to modulate swing speed without relying on feedback information. This control mode basically functions as open-loop control.

**Intelligent mode.**    Conversely, the intelligent mode, recently proposed in works as six neural CPG-based control modules with interconnected feedback [13]. The intelligent control mechanism, acting as adaptive closed-loop control, provided gait generation and adaptation in response to the tracking error during locomotion. Key components of six neural control modules include phase generator, pattern generator, transformation equation (forward dynamics), time constant, gradient-based adaptation, and coupled cooperative primitives (CCP) adaptation. The phase generator creates rhythmic phase signals ($C_1$, $C_2$) based on a central pattern generator (CPG) concept [21].

Full descriptions of Exo-H3 hardware and the development of default and intelligent modes are available in the supplementary material (S1 File). Note that the Exo-H3 functions as an adaptive exoskeleton when using the intelligent control mode, and as a non-adaptive exoskeleton with the default mode. The design parameters affected the control performance in both speeds were detailed and illustrated in the supplementary file (S1 File). The parameters were the same for both speeds. Only 10% incremental of the preferred speed was set as the high speed. The set of weights used to generate joint trajectories (hip, knee, and ankle) were the same for both speeds. The pattern generator block has learnt the weights from the default patterns provided by the manufacturer of Exo-H3 during the offline process. The same pattern was also applied for both speeds during the default mode. This might result in slightly different patterns which were still acceptable for guiding the subjects. For phase generator where CPG was deployed, all parameters were tuned to be able to generate certain swing frequency range (0 < swing freq. < 1 Hz). This range covered all low and high speeds in our experiments.

## 2.6 Data collection

All participants were initially instructed to walk at their preferred speed without wearing an exoskeleton. Subsequently, individuals walked with the assigned exoskeleton at normal speed and then at high speed. The participants had a minimum of 10 minutes of rest [23], or more if needed, between each intervention to minimize carryover effects [24]. Before starting the experiment, each participant underwent training to walk using the default-mode exoskeleton. They were instructed to choose comfortable walking speed, and high-speed walking was set at 10% of their preferred speed. For the intelligent mode, the same walking speed was applied as that of the default settings. Details for subjects' speed were outlined in S2 File. All participants were evaluated spatio-temporal parameters, pelvic-hip-knee-ankle kinematics, muscle activity, the maximum rate of oxygen consumption ($VO_2$max), comfort score and performance of the exoskeleton user on the same day as the experiment.

Kinematics and spatio-temporal parameters were collected by using three dimensions with an eight-camera Motion analysis (Motion Analysis Corporation, Santa Rosa, CA, USA). Twenty-nine reflective markers based on modified Helen Hayes were used for tracking joint movements (S3 File). In cases where the marker positions were obscured by the exoskeleton, the researcher placed markers on the exoskeleton as close to the body part as possible, with regard to the Helen Hayes marker set. Participants walked back and forth in the straight line 8 meters for 10–15 round. Muscle activity was assessed by surface electromyography (sEMG),

ProEMG software at 2000 Hz, and a Myon 320 wireless EMG (Myon AG, Schwarzenberg, Switzerland). All EMG data were processed using the standard filtered and rectified method, a 10–450 Hz band pass filter. For $VO_2$ max, participants performed the 6-minute walk test (6MWT) which recorded by Oxycon Mobile (Erich Jaeger GmbH, Hoechberg, Germany). To ensure the accuracy and reliability of our measurements, we performed calibration of these systems before each data collection session. After finished the walking test, the user comfortability and the performance of the exoskeleton control were assessed using developed questionnaires according to comfort rating scales for wearable devices and people health [25, 26] and the modified questionnaire form (from WOMAC) in Thai version for assessment [27], respectively.

## 2.7 Outcomes

The primary outcome measure of the study was kinematics of gait, while the secondary outcome measures were spatio-temporal parameters, muscle activity, oxygen consumption, user comfort, and exoskeleton performance scores.

## 2.8 Data analysis

Gait, kinematics, and spatio-temporal parameters were quantified in the stance and swing phases of gait cycle. The individual's data were selected for 3 walking trials and averaged, and each trial was selected 1–2 gait cycle. Mean velocity, cadence, step length, step width, stride length, stance phase, swing phase time, bilateral pelvic-hip-knee-ankle angles (degrees) in the frontal, sagittal and transverse plane were collected. Pelvic obliquity, pelvic tilt, pelvic rotation, hip abduction/adduction, hip flexion/extension, hip rotation, knee valgus/varus, knee flexion/extension, knee rotation, ankle inversion/eversion, dorsiflexion-plantar flexion, foot progression, and also ankle rotation was assessed by motion analysis software (Orthotrak, Motion Analysis Corp). The kinematics data were captured according to 0–100% of the gait cycle at heel strike (0%–2%), midstance (12%–31%), terminal stance (31%–50%), and mid-swing (74%–87%) [28].

Regarding to walking, bilateral muscles activities during walking were quantified as maximum voluntary contraction (MVC) and the relative percentage of root mean square (%RMS) value of gluteus medius, gluteus maximus, tensor fascia latae, rectus femoris, bicep femoris, medial gastrocnemius, and tibialis anterior muscles for the walking trial. Muscle activities (voltage; V), and averaged sEMG activity (%MVC) were reported using ProEMG 2.1 software (Prophysics AG, Schaffhauserstrasse, Kloten, Switzerland). $VO_2$ MAX (ml/kg/min) was estimated by Lab start-up 5.0 software.

For the wearer's opinion to the device, their perceived change, emotion on self-image, anxiety on security, harm or painful, attachment, and movement were evaluated as the comfort score [25]. Each item rated from a 1 (the least problem) to 5 (the most problem). Sum of total score was 30, and categorized as very comfortable (1–6 points), comfortable (7–17 points), moderately comfortable (18 points), uncomfortable (19–29 points), and uncomfortable at all (30 points).

User experience was evaluated using an exoskeletal performance questionnaire that comprised two domains: pain (while walking, rising from sitting, and standing) and difficulty (in rising from sitting, standing, forward bending, walking, and walking for distance). In accordance with our protocol focused on gait measurement, this approach captured the most relevant experiences of discomfort (pain) and difficulty, providing a comprehensive assessment of the user experience with the exoskeleton. Respondents rated all 8 questions ranged from 1 (minimal issues) to 5 (significant problems). The total performance score was designated as

the best performance (8 points), good performance (9–23 points), moderate performance (24 points), poor performance (25–39 points), and the worst performance (40 points).

### 2.9 Sample size

Sample size estimation was conducted using a sample size calculator for crossover design (https://www2.ccrb.cuhk.edu.hk/stat/mean/tsmc_sup.htm). The estimation was based on a mean difference of 6.8 degrees [29], a standard deviation (SD) of 7.6 degrees, and a margin of 5 degrees. The study used a type I error (alpha) of 0.05 and power of 0.8. Therefore, the total sample size required for the study was 8 participants.

### 2.10 Statistical analysis

The Shapiro-Wilk test was test for normality of data. Descriptive data was presented as mean, standard deviation, frequency and percentage. Differences of demographic data, temporal spatial data, kinematics, and muscle activity between two assigned groups were analyzed using unpaired t-test and Chi-square test. To compare gait parameters, muscle activities, $VO_2$ max, comfortability and performance scores between exoskeleton modes, paired t-test was used for normally distributed data, otherwise Wilcoxon signed rank test was applied. Statistical analysis was performed using Stata software version 15.0, (Stata Corp, College Station, Texas, USA). The significance level was set at P < 0.05.

Statistical parametric mapping (SPM) was conducted to compare time series of joint kinematics. SPM paired t-tests were used to compare the mean kinematics of each joint movement between exoskeleton mode and normal walking, as well as between two exoskeleton modes at normal and high speed. The analysis was performed using MATLAB (R2020b, The MathWorks Inc) and open-source SPM1d code (version M.0.4.10, www.spm1D.org), with a significance level set at alpha error = 0.05.

## 3. Results

Eight participants (3 males and 5 females) were recruited and completed the trial. (See flowchart, Fig 1). Their average age was 25.8 years (range 20–30 years). Baseline characteristics was shown in S1 Table. Group 1 had significantly lower mean weight (51.25±2.50 kg) and BMI (18.82±0.35 kg/m²) compared to Group 2 (weight: 65.25±7.27 kg; BMI: 23.38±2.21 kg/m²), Table 1. Age, sex, height, shoe size, underlying disease, education, temporal spatial, and $VO_2$ max were insignificant differences between groups.

According to baseline normal walking kinematics, Group 1 exhibited lower mean hip flexion than Group 2 at heel strike (24.11 ± 6.84 vs 34.53 ± 5.01 degrees, p = 0.049) and mid-stance (9.33, ± 5.18 vs 19.16 ± 4.84, p = 0.032). Other kinematic data did not reach significant differences, S2 Table.

The muscle activity during baseline walking was displayed in S3 Table. Group 1 demonstrated significantly lower %RMS of the right biceps femoris when compared to Group 2 (8.55 ± 4.01 vs 23.58±11.39 voltage, p = 0.047). No significant difference between groups was found in other muscles, including the gluteus maximus, tibialis anterior, gastrocnemius, and rectus femoris.

The temporal spatial between walking with the exoskeleton at different speed was compared between the intelligent and default modes (Table 2). The former showed significantly lesser velocity and stride step lengths than the latter at both patient's preferred walking and high speed. Walking with and without the exoskeleton in different modes and speed explicitly contrasted. Regarding normal-speed walking, the default mode (NA) significantly compromised velocity, cadence, and %left swing with longer step width. Whereas the intelligent mode (NB)

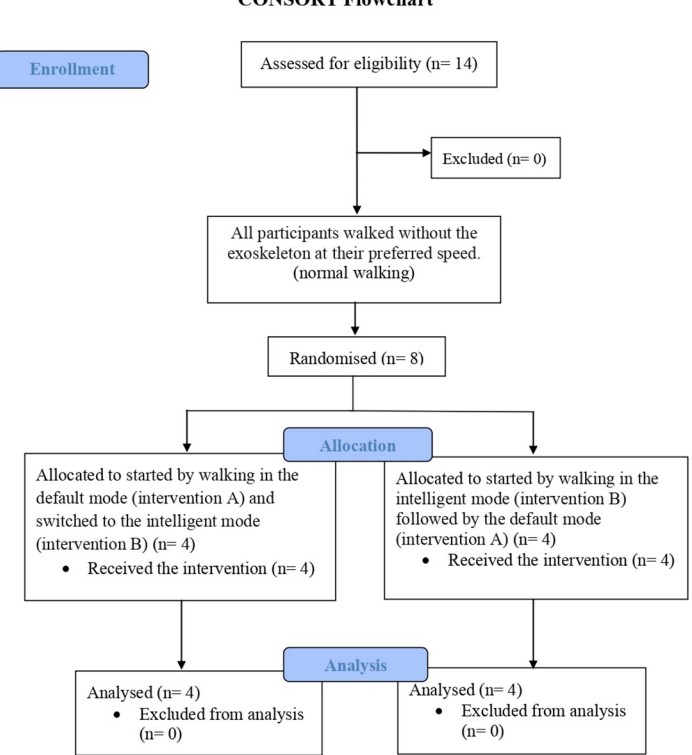

**Fig 1. Flowchart of the study population.**

additionally diminished stride, step lengths, and swing phase. Both exoskeleton modes at high speed (HA, HB) was also significant difference from the normal walking (NW) in velocity, cadence, step width, percentage of stance and swing time. The high-speed modes were slightly closed to the normal walking than the normal-speed modes.

Pelvic-hip-knee-ankle-foot kinematics of the exoskeleton (intelligent and default modes), and normal walking were showed in Table 3. Comparing to the default setting, the intelligent-mode exoskeleton significantly increased anterior pelvic tilt during midstance and mid-swing in both normal speed (2.70 ±2.91 degrees, p = 0.034 and 3.69 ± 1.77 degrees, p = 0.001) and high speed (1.41 ± 1.15 degrees, p = 0.011 and 2.52 ± 1.69 degrees, p = 0.004). It also provided more pelvic internal rotation during heel strike (p = 0.009), pelvic external rotation during mid-swing (p = 0.003) at normal speed and pelvic internal rotation during heel strike at high speed (p = 0.006).

Comparing hip kinematics to the default, the intelligent mode significantly decreased hip adduction by -2.25 ±1.53 degrees during mid-swing at normal speed and increased hip adduction by 2.24 ±2.09 degrees during heel strike at high speed. This setting also limited hip flexion during heel strike at normal speed (-4.44 ± 4.62 degrees), at high speed (-6.90 ±6.12 degrees), and during mid-swing at normal speed (-2.96±2.93 degrees). On the other hand, the intelligent mode allowed more hip flexion during the stance phase at normal speed. These ranges were closed to normal walking than the default ones.

In regards to the default, the intelligent mode decreased knee varus at midstance, and accelerated varus at mid-swing in both speeds. This modality also enhanced knee flexion during stance phase but diminished knee flexion during mid-swing. Additionally, it provided more

**Table 1. Demographic and baseline characteristics.**

| Demographic data and temporal spatial data | Group 1 (n = 4) | Group 2 (n = 4) | P-value |
|---|---|---|---|
| **Age (year),** mean (SD) | 24.75 (4.03) | 27 (4.08) | 0.463 |
| **Sex**, n (%) | | | 0.465 |
| Male | 2 (50.00) | 1 (25.00) | |
| Female | 2 (50.00) | 3 (75.00) | |
| **Weight (kg),** mean (SD) | 51.25 (2.50) | 65.25 (7.27) | 0.011* |
| **Height (cm),** mean (SD) | 165 (4.16) | 167 (4.97) | 0.559 |
| **BMI (kg/m$^2$),** mean (SD) | 18.82 (0.35) | 23.38 (2.21) | 0.007* |
| **Shoes size (EU),** mean (SD) | 39.75 (2.22) | 39.25 (1.26) | 0.708 |
| **Underlying disease,** n (%) | | | - |
| Yes | 0 (0.00) | 0 (0.00) | |
| No | 4 (100.00) | 4 (100.00) | |
| **Education,** n (%) | | | 0.285 |
| High school | 1(25.00) | 0 (0.00) | |
| Bachelor | 3 (75.00) | 4 (100.00) | |
| **Velocity (cm/s),** mean (SD) | 97.05 (6.09) | 103.85 (14.34) | 0.416 |
| **Cadence (step/min),** mean (SD) | 102.38 (7.81) | 106.3 (5.48) | 0.442 |
| **Stride length (cm),** mean (SD) | 116.8 (7.13) | 117.85 (9.81) | 0.868 |
| **Step width (cm),** mean (SD) | 9.40 (3.03) | 12.20 (3.49) | 0.271 |
| **Right step length (cm),** mean (SD) | 69.23 (19.89) | 51.05 (18.42) | 0.229 |
| **Left step length (cm),** mean (SD) | 47.80 (16.94) | 66.4 (13.54) | 0.137 |
| **Right stance phase (%),** mean (SD) | 62.60 (1.59) | 60.90 (1.59) | 0.181 |
| **Left stance phase (%),** mean (SD) | 59.35 (5.42) | 61.68 (1.56) | 0.441 |
| **Right swing phase (%),** mean (SD) | 37.40 (1.59) | 39.10 (1.59) | 0.181 |
| **Left swing phase (%),** mean (SD) | 40.65 (5.42) | 38.33 (1.56) | 0.441 |
| **VO$_2$ max (ml/kg/min),** mean (SD) | 18.18 (4.65) | 13.05 (2.87) | 0.109 |

BMI = body mass index, SD = standard deviation,

* significant P-value $<0.05$

ankle varus and dorsiflexion at normal speed as well as ankle internal rotation (4.94 degrees) during mid-swing, internal rotation of foot progression during heel strike, external rotation of foot progression during mid-swing at high speed than the default-mode exoskeleton.

Kinematics of bimodal exoskeletons were significant differences from those of normal walking (Table 3). Comparing with normal walking at normal speed, the intelligent and the default modes showed more downward pelvic obliquity at terminal stance to mid-swing, posteriorly tilt (Fig 2), hip extension, and more hip abduction (Fig 3). Bimodal settings significantly increased hip extension, and abduction throughout gait cycle in regards with both speeds. Moreover, they produced more varus, extension, and internal rotation of the knees (Fig 4) as well as ankle varus, plantar flexion at terminal stance, ankle internal rotation and internal foot progression (Fig 5). Similar kinematic patterns were found at high speed.

SPM paired t-test indicated insignificant differences of pelvic, hip and ankle angles between the intelligent and the default mode at various speed. Compared to normal walking, the exoskeleton at both speeds significantly increased posterior pelvic tilt during heel strike, terminal stance, and mid-swing (Fig 6). The intelligent mode at both speeds produced more pelvic downward during 30–40% of gait cycle and upward obliquity during terminal swing (p<0.001). The exoskeleton markedly increased hip abduction, extension and external

**Table 2. The comparison of the temporal spatial between the exoskeleton mode and normal walking under different speed.**

| Temporal spatial | Normal speed | | | | | | High speed | | | | | |
|---|---|---|---|---|---|---|---|---|---|---|---|---|
| | NA vs NW | | NB vs NW | | NA vs NB | | HA vs NW | | HB vs NW | | HA vs HB | |
| | Mean difference (SD) | P-value | Mean difference (SD) | P-value | Mean difference (SD) | P-value | Mean difference (SD) | P-value | Mean difference (SD) | P-value | Mean difference (SD) | P-value |
| Velocity (cm/s) | -62.84 (12.41) | <0.001* | -72.28 (11.02) | <0.001* | -9.44 (6.76) | 0.006* | -57.96 (14.31) | <0.001* | -66.86 (14.21) | <0.001* | -8.90 (8.39) | 0.019* |
| Cadence (step/min) | -63.93 (7.78) | <0.001* | -65.74 (7.66) | <0.001* | -1.81 (2.39) | 0.069 | -60.73 (7.82) | <0.001* | -59.85 (11.05) | <0.001* | 0.88 (7.15) | 0.739 |
| Stride length (cm) | -5.20 (17.08) | 0.418 | -28.76 (14.35) | <0.001* | -23.56 (19.65) | 0.012* | 0.30 (18.13) | 0.964 | -21.89 (30.42) | 0.081 | -22.19 (22.94) | 0.029* |
| Step width (cm) | 18.26 (4.36) | <0.001* | 18.66 (2.44) | <0.001* | 0.4 (4.25) | 0.798 | 17.35 (4.62) | <0.001* | 18.04 (3.36) | <0.001* | 0.69 (4.30) | 0.665 |
| Right step length (cm) | -1.75 (21.15) | 0.822 | -12.69 (22.55) | 0.156 | -10.94 (11.03) | 0.026* | -0.06 (23.65) | 0.994 | -8.60 (27.09) | 0.399 | -8.54 (11.74) | 0.079 |
| Left step length (cm) | -2.48 (19.54) | 0.731 | -16.00 (18.16) | 0.042* | -13.53 (9.3) | 0.005* | -0.68 (17.26) | 0.915 | -13.94 (20.58) | 0.097 | -13.26 (13.26) | 0.025* |
| % Right stance | 4.19 (12.26) | 0.366 | 8.56 (3.14) | <0.001* | 4.38 (12.65) | 0.360 | 8.39 (2.69) | <0.001* | 7.56 (4.12) | 0.001* | -0.83 (3.76) | 0.555 |
| % Left stance | 9.06 (3.87) | <0.001* | 10.09 (5.68) | 0.002* | 1.03 (2.27) | 0.243 | 9.05 (5.83) | 0.003* | 10.03 (5.66) | 0.002* | 0.98 (3.74) | 0.485 |
| % Right swing | -4.19 (12.26) | 0.366 | -8.56 (3.14) | <0.001* | -4.38 (12.65) | 0.360 | -8.39 (2.69) | <0.001* | -7.56 (4.12) | 0.001* | 0.83 (3.76) | 0.555 |
| % Left swing | -9.06 (3.87) | <0.001* | -10.09 (5.68) | 0.002* | -1.03 (2.27) | 0.243 | -9.05 (5.83) | 0.003* | -10.03 (5.66) | 0.002* | -0.98 (3.74) | 0.485 |

NA = normal-speed walking with the default-mode exoskeleton, NW = normal walking, NB = normal-speed walking with the intelligent-mode exoskeleton, HA = high-speed walking with the default-mode exoskeleton, HB = high-speed walking with the intelligent-mode exoskeleton, SD = standard deviation,

*significant P-value <0.05 from paired-t test.

rotation almost 90% of gait cycle when compared to normal walking (Fig 7). The knee flexion of each mode significantly contrasted during 75–85% of gait cycle at normal speed (p = 0.003) and high speed (p = 0.019) (Fig 8). With respect to normal walking, both exoskeletal modes revealed similar knee patterns, but visibly increased ankle varus across two speed (p<0.001), Fig 9. The intelligent mode generated more ankle internal rotation, whereas the default significantly increased plantar flexion during mid-stance (p = 0.004) at normal speed.

Muscle activity, $VO_2$ max, the user's comfort score and exoskeleton performance score between walking with an exoskeleton and normal walking under different modes and speed were reported in Table 4. Both modes demonstrated indifferent muscle activity. Comparing to the normal walking, the default-mode exoskeleton significantly contributed to higher muscle activities, i.e., left gastrocnemius (122.23±190.84 V) at normal speed and left biceps femoris (21.42±43.16 V) at high speed. The default setting also required higher $VO_2$ max than the intelligent mode, and normal walking. Whereas the intelligent mode and normal walking indifferently consumed $VO_2$ max at both speeds. The user's comfort score and the exoskeleton performance score did not reach significant different between both modalities.

## 4. Discussion

This crossover study compared the performance of a state-of-the-art adaptive CPG-based control system (intelligent mode) to a standard commercial control system (default mode) in a lower-limb exoskeleton. We evaluated various metrics including temporal-spatial gait

**Table 3. The comparisons of kinematics data between the exoskeleton modes and normal walking.**

| Kinematics | Normal speed | | | | | | High speed | | | | | |
|---|---|---|---|---|---|---|---|---|---|---|---|---|
| | NA vs NW | | NB vs NW | | NB vs NA | | HA vs NW | | HB vs NW | | HB vs HA | |
| | Mean difference (SD) | P-value | Mean difference (SD) | P-value | Mean difference (SD) | P-value | Mean difference (SD) | P-value | Mean difference (SD) | P-value | Mean difference (SD) | P-value |
| **Pelvis** | | | | | | | | | | | | |
| **Pelvic Obliquity Up/Down** | | | | | | | | | | | | |
| Heel strike | -0.41 (1.94) | 0.573 | 1.20 (2.75) | 0.256 | 1.61 (2.62) | 0.126 | -1.43 (1.98) | 0.080 | 0.74 (2.77) | 0.471 | 2.18 (2.23) | 0.028* |
| Mid-stance | -2.76 (3.52) | 0.062 | -2.04 (2.42) | 0.049* | 0.72 (1.59) | 0.246 | -1.66 (1.61) | 0.022* | -0.65 (1.82) | 0.343 | 1.01 (1.25) | 0.055 |
| Terminal stance | -2.08 (2.05) | 0.025*[a] | -3.58 (1.98) | 0.012*[a] | -1.49 (2.39) | 0.120 | -0.95 (1.19) | 0.093[a] | -2.34 (1.44) | 0.012*[a] | -1.38 (1.21) | 0.014* |
| Mid swing | 4.33 (3.89) | 0.006* | 4.11 (2.97) | 0.006* | -0.22 (1.85) | 0.748 | 3.07 (0.72) | <0.001* | 2.72 (1.44) | 0.001* | -0.35 (1.32) | 0.472 |
| **Pelvic Tilt Anterior/Posterior** | | | | | | | | | | | | |
| Heel strike | -9.34 (7.62) | 0.010* | -10.15 (8.19) | 0.010* | -0.81 (0.53) | 0.532 | -9.61 (8.52) | 0.015* | -11.75 (7.29) | 0.003* | -2.15 (1.99) | 0.019* |
| Mid-stance | -6.89 (7.79) | 0.041* | -4.19 (7.98) | 0.181 | 2.70 (2.91) | 0.034* | -7.37 (8.06) | 0.036* | -5.96 (7.64) | 0.063 | 1.41 (1.15) | 0.011* |
| Terminal stance | -10.38 (8.19) | 0.009* | -9.67 (8.44) | 0.014* | 0.70 (2.29) | 0.415 | -11.22 (8.4) | 0.007* | -11.30 (7.19) | 0.003* | -0.07 (1.79) | 0.912 |
| Mid swing | -9.45 (7.91) | 0.012* | -5.76 (8.63) | 0.101 | 3.69 (1.77) | 0.001* | -9.89 (8.32) | 0.012* | -7.38 (7.49) | 0.027* | 2.52 (1.69) | 0.004* |
| **Pelvic Internal / External Rotation** | | | | | | | | | | | | |
| Heel strike | -0.44 (2.99) | 0.691 | 2.36 (2.93) | 0.057 | 2.79 (2.22) | 0.009* | -0.59 (3.52) | 0.647 | 3.12 (2.82) | 0.017* | 3.72 (2.68) | 0.006* |
| Mid-stance | -1.88 (3.56) | 0.180 | 0.95 (3.93) | 0.516 | 2.83 (3.53) | 0.058 | -1.83 (3.57) | 0.189 | 1.00 (5.46) | 0.619 | 2.84 (5.24) | 0.170 |
| Terminal stance | 1.86 (2.44) | 0.068 | 1.54 (2.04) | 0.069 | -0.31 (1.47) | 0.563 | 1.11 (3.39) | 0.387 | 1.51 (3.19) | 0.224 | 0.39 (3.72) | 0.770 |
| Mid swing | 1.81 (2.98) | 0.129 | -2.15 (2.87) | 0.072 | -3.96 (2.51) | 0.003* | 1.27 (3.28) | 0.208[a] | -1.52 (7.49) | 0.779[a] | -2.79 (5.14) | 0.161[a] |
| **Hip** | | | | | | | | | | | | |
| **Hip Adduction /Abduction** | | | | | | | | | | | | |
| Heel strike | -16.09 (4.27) | <0.001* | -14.79 (4.69) | 0.012*[a] | 1.30 (2.18) | 0.135 | -16.30 (4.43) | <0.001* | -14.06 (4.92) | 0.0001* | 2.24 (2.09) | 0.019* |
| Mid-stance | -15.12 (3.89) | <0.001* | -14.49 (3.11) | <0.001* | 0.62 (1.80) | 0.363 | -14.00 (2.50) | <0.001* | -12.96 (3.23) | <0.001* | 1.04 (1.85) | 0.154 |
| Terminal stance | -12.75 (3.25) | <0.001* | -14.74 (3.29) | <0.001* | -1.99 (3.33) | 0.135 | -11.91 (1.84) | <0.001* | -13.75 (2.69) | <0.001* | -1.84 (2.24) | 0.053 |
| Mid swing | -9.05 (4.07) | <0.001* | -11.30 (3.54) | <0.001* | -2.25 (1.53) | 0.004* | -10.06 (3.21) | <0.001* | -11.43 (3.09) | <0.001* | -1.37 (2.07) | 0.103 |
| **Hip Flexion/ Extension** | | | | | | | | | | | | |
| Heel strike | -11.89 (12.09) | 0.027* | -16.34 (12.29) | 0.007* | -4.44 (4.62) | 0.030* | -11.07 (11.46) | 0.029* | -17.97 (10.09) | 0.002* | -6.90 (6.12) | 0.015* |
| Mid-stance | -15.70 (11.28) | 0.006* | -12.89 (11.31) | 0.015* | 2.81 (2.68) | 0.021* | -15.51 (10.85) | 0.005* | -14.01 (10.36) | 0.007* | 1.49 (2.29) | 0.108 |
| Terminal stance | -15.26 (10.81) | 0.005* | -11.61 (10.87) | 0.019* | 3.65 (3.08) | 0.012* | -15.91 (10.41) | 0.004 * | -13.28 (10.44) | 0.009* | 2.64 (2.01) | 0.008* |
| Mid swing | -27.49 (9.43) | <0.001* | -30.45 (9.79) | <0.001* | -2.96 (2.93) | 0.024* | -28.54 (9.58) | <0.001* | -31.21 (7.49) | <0.001* | -2.67 (3.96) | 0.098 |
| **Hip Internal / External Rotation** | | | | | | | | | | | | |

(*Continued*)

**Table 3.** (Continued)

| Kinematics | Normal speed | | | | | | High speed | | | | | |
|---|---|---|---|---|---|---|---|---|---|---|---|---|
| | NA vs NW | | NB vs NW | | NB vs NA | | HA vs NW | | HB vs NW | | HB vs HA | |
| | Mean difference (SD) | P-value | Mean difference (SD) | P-value | Mean difference (SD) | P-value | Mean difference (SD) | P-value | Mean difference (SD) | P-value | Mean difference (SD) | P-value |
| Heel strike | -14.91 (8.98) | 0.002* | -14.34 (9.68) | 0.004* | 0.57 (3.15) | 0.623 | -15.19 (8.64) | 0.002* | -14.83 (8.36) | 0.002* | 0.35 (1.92) | 0.619 |
| Mid-stance | -10.00 (9.78) | 0.023* | -10.75 (10.29) | 0.021* | -0.75 (2.46) | 0.419 | -10.98 (9.57) | 0.014* | -11.29 (10.71) | 0.021* | -0.31 (2.43) | 0.730 |
| Terminal stance | -9.25 (9.49) | 0.028* | -8.02 (9.39) | 0.046* | 1.23 (2.11) | 0.144 | -9.57 (9.09) | 0.021* | -8.37 (10.39) | 0.056* | 1.19 (2.68) | 0.247 |
| Mid swing | -14.03 (9.71) | 0.005* | -14.15 (10.20) | 0.006* | -0.12 (2.44) | 0.892 | -12.61 (9.03) | 0.006* | -12.52 (8.96) | 0.006* | -1.54 (2.67) | 0.148 |
| **Knee** | | | | | | | | | | | | |
| **Knee Varus/Valgus** | | | | | | | | | | | | |
| Heel strike | 11.44 (1.99) | <0.001* | 10.53 (2.55) | <0.001* | -0.91 (1.23) | 0.075 | 11.50 (2.04) | <0.001* | 10.12 (1.84) | <0.001* | -1.38 (0.89) | 0.003* |
| Mid-stance | 13.02 (2.26) | 0.012*a | 11.90 (3.41) | <0.001* | -1.12 (1.33) | 0.049* | 13.06 (2.42) | <0.001* | 11.87 (2.71) | <0.001* | -1.19 (1.31) | 0.036* |
| Terminal stance | 11.88 (1.59) | 0.012*a | 11.66 (1.64) | 0.012*a | -0.22 (0.93) | 0.523 | 12.11 (1.50) | 0.012*a | 11.83 (1.90) | 0.012*a | -0.28 (0.91) | 0.401a |
| Mid swing | -0.05 (5.78) | 0.982 | 4.72 (4.98) | 0.032* | 4.77 (2.48) | 0.001* | 0.55 (5.94) | 0.800 | 4.49 (6.49) | 0.091 | 3.93 (4.01) | 0.027* |
| **Knee Flexion/Extension** | | | | | | | | | | | | |
| Heel strike | -3.09 (5.58) | 0.162 | 2.09 (6.12) | 0.366 | 5.18 (4.17) | 0.010* | -3.58 (5.54) | 0.109 | -1.07 (6.60) | 0.779 a | 2.51 (4.36) | 0.161a |
| Mid-stance | -11.8 (7.28) | 0.003* | -7.93 (8.08) | 0.028* | 3.89 (3.56) | 0.018* | -11.74 (7.18) | 0.002* | -8.96 (7.72) | 0.014* | 2.79 (2.74) | 0.024* |
| Terminal stance | -10.17 (5.96) | 0.002* | -5.28 (6.45) | 0.054 | 4.89 (3.07) | 0.003* | -9.99 (6.27) | 0.003* | -6.42 (7.24) | 0.041* | 3.58 (2.95) | 0.011* |
| Mid swing | -1.58 (15.68) | 0.161a | -16.67 (13.72) | 0.025*a | -15.08 (6.84) | 0.012*a | -2.72 (14.26) | 0.161a | -17.51 (16.20) | 0.025*a | -14.79 (8.92) | 0.002* |
| **Knee Internal /External Rotation** | | | | | | | | | | | | |
| Heel strike | 22.40 (12.62) | 0.002* | 22.24 (12.47) | 0.002* | -0.16 (3.13) | 0.891 | 23.34 (11.68) | 0.001* | 20.73 (14.93) | 0.006* | -2.61 (3.70) | 0.086 |
| Mid-stance | 14.29 (12.30) | 0.013* | 14.74 (11.63) | 0.009* | 0.45 (3.52) | 0.731 | 14.59 (11.72) | 0.010* | 13.54 (12.56) | 0.019* | -1.06 (2.41) | 0.256 |
| Terminal stance | 11.85 (11.03) | 0.036*a | 13.54 (10.80) | 0.009* | 1.69 (3.13) | 0.124a | 12.59 (10.89) | 0.014* | 12.17 (11.47) | 0.020* | -0.42 (1.89) | 0.554 |
| Mid swing | 34.57 (10.15) | <0.001* | 34.46 (11.57) | 0.0001* | -0.12 (2.62) | 0.902 | 36.24 (10.71) | <0.001* | 32.85 (11.91) | 0.0001* | -3.39 (3.25) | 0.021* |
| **Ankle** | | | | | | | | | | | | |
| **Ankle Varus/Valgus** | | | | | | | | | | | | |
| Heel strike | 66.68 (10.02) | <0.001* | 69.14 (11.09) | 0.012*a | 2.46 (2.35) | 0.021* | 68.76 (9.76) | <0.001* | 69.09 (10.56) | 0.012*a | 0.33 (3.02) | 0.674 a |
| Mid-stance | 67.58 (11.29) | <0.001* | 69.28 (12.07) | <0.001* | 1.69 (1.20) | 0.005* | 68.63 (10.91) | <0.001* | 69.86 (11.81) | 0.012*a | 1.23 (2.09) | 0.208a |
| Terminal stance | 71.67 (13.32) | <0.001* | 72.31 (13.96) | <0.001* | 0.64 (1.03) | 0.122 | 71.61 (13.04) | <0.001* | 72.72 (13.79) | 0.012*a | 1.11 (1.28) | 0.050a |
| Mid swing | 74.78 (10.51) | 0.012*a | 74.05 (10.27) | 0.012* a | -0.73 (2.16) | 0.484 a | 74.37 (10.72) | <0.001* | 74.89 (9.79) | <0.001* | 0.52 (2.15) | 0.514 |
| **Ankle Dorsiflexion/Plantar flexion** | | | | | | | | | | | | |
| Heel strike | -7.97 (14.35) | 0.160 | -4.68 (11.46) | 0.286 | 3.29 (4.41) | 0.072 | -5.09 (4.55) | 0.332 | -4.49 (14.51) | 0.410 | 0.60 (8.57) | 0.849 |

(*Continued*)

**Table 3.** (Continued)

| Kinematics | Normal speed | | | | | | High speed | | | | | |
|---|---|---|---|---|---|---|---|---|---|---|---|---|
| | NA vs NW | | NB vs NW | | NB vs NA | | HA vs NW | | HB vs NW | | HB vs HA | |
| | Mean difference (SD) | P-value | Mean difference (SD) | P-value | Mean difference (SD) | P-value | Mean difference (SD) | P-value | Mean difference (SD) | P-value | Mean difference (SD) | P-value |
| Mid-stance | -11.33 (7.89) | 0.005* | -7.47 (7.55) | 0.027 | 3.86 (3.09) | 0.010* | -9.68 (7.96) | 0.011* | -9.00 (7.68) | 0.013* | 0.68 (7.87) | 0.814 |
| Terminal stance | -12.24 (11.52) | 0.020* | -9.21 (10.39) | 0.041* | 3.03 (3.01) | 0.025* | -9.57 (11.39) | 0.049* | -9.91 (13.01) | 0.068 | -0.34 (5.34) | 0.863 |
| Mid swing | -13.90 (20.76) | 0.124[a] | -8.97 (21.08) | 0.268 | 4.94 (6.92) | 0.093[a] | -4.71 (15.53) | 1.000 | -7.36 (21.17) | 0.575[a] | -2.65 (8.19) | 0.401[a] |
| **Ankle Internal/External Rotation** | | | | | | | | | | | | |
| Heel strike | 13.05 (9.59) | 0.006* | 12.45 (6.92) | 0.001* | -0.59 (3.70) | 0.662 | 10.24 (9.05) | 0.015* | 11.46 (13.63) | 0.049 | 1.21 (7.19) | 0.649 |
| Mid-stance | 10.99 (7.99) | 0.006* | 9.96 (6.66) | 0.004* | -1.03 (2.86) | 0.342 | 8.56 (6.81) | 0.009* | 9.54 (9.60) | 0.026* | 1.24 (3.95) | 0.499 |
| Terminal stance | 12.73 (9.87) | 0.008* | 10.67 (9.61) | 0.016* | -2.06 (2.71) | 0.069 | 9.86 (9.18) | 0.019* | 10.41 (11.98) | 0.044* | 0.54 (4.75) | 0.756 |
| Mid swing | 14.94 (19.04) | 0.069 | 16.02 (19.47) | 0.012*[a] | 1.08 (6.71) | 0.662 | 7.58 (16.84) | 0.161 | 12.52 (20.06) | 0.124 | 4.94 (4.55) | 0.036*[a] |
| **Internal/External Foot Progression** | | | | | | | | | | | | |
| Heel strike | 9.87 (4.78) | 0.001* | 11.28 (8.34) | 0.0001* | 1.42 (2.07) | 0.093 | 8.32 (4.53) | 0.001* | 10.03 (4.39) | 0.0003* | 1.71 (1.02) | 0.002* |
| Mid-stance | 4.57 (3.45) | 0.007* | 5.51 (2.62) | 0.001* | 0.94 (2.09) | 0.245 | 2.34 (3.33) | 0.088 | 3.58 (4.59) | 0.064 | 1.24 (3.95) | 0.402 |
| Terminal stance | 6.42 (3.09) | 0.001* | 7.57 (2.54) | 0.0001* | 1.16 (1.60) | 0.081 | 4.31 (3.53) | 0.011* | 5.51 (4.63) | 0.012* | 1.20 (3.99) | 0.423 |
| Mid swing | 18.70 (6.77) | 0.0001* | 13.45 (7.93) | 0.002* | -5.25 (2.69) | 0.001* | 18.51 (6.24) | 0.0001* | 14.14 (8.36) | 0.002* | -4.38 (4.82) | 0.037* |

NA = normal-speed walking with the default-mode exoskeleton, NW = normal walking, NB, normal-speed walking with the intelligent-mode exoskeleton, HA = high-speed walking with the default-mode exoskeleton, HB = high-speed walking with the intelligent-mode exoskeleton, SD = standard deviation, + = pelvic obliquity up, pelvic tilt anterior, pelvic internal rotation, hip adduction, hip flexion, hip internal rotation, knee valgus, knee flexion, knee internal rotation, ankle varus, ankle dorsiflexion, ankle internal rotation, foot internal rotation,— = pelvic obliquity down, pelvic tilt posterior, pelvic external rotation, hip abduction, hip extension, hip external rotation, knee valgus, knee extension, knee external rotation, ankle valgus, ankle plantar flexion, ankle external rotation, foot external rotation,

*P-value <0.05, P-value from Paired t-test,

[a]P-value from Wilcoxon signed-rank test.

characteristics, kinematics, muscle activity, oxygen consumption, user comfort, and exoskeleton performance. In regard to the default, the intelligent modification compromised velocity, stride and step lengths, but allowed more anterior pelvic tilt, pelvic rotation, knee varus, ankle varus, ankle internal rotation, and external rotation of foot progression during mid-swing. At high-speed walking, this intelligent mode slightly closed to the normal kinematics than the default. Muscle activity, comfort and performance score was comparable between modalities. However, the default-mode exoskeleton significantly increased left gastrocnemius and left biceps femoris when compared to normal walking. The default setting also required higher $VO_2$ max than the intelligent mode, and normal gait.

The exoskeleton specifically changed gait parameters and walking habits in order to maintain the dynamic balance of the human body [30–32]. In many commercialized exoskeletons such as ReWalk, Ekso, etc., they usually adopt the basic joint trajectory tracking control as their default control [11]. The adaptive control, as in our case, is considered to be more

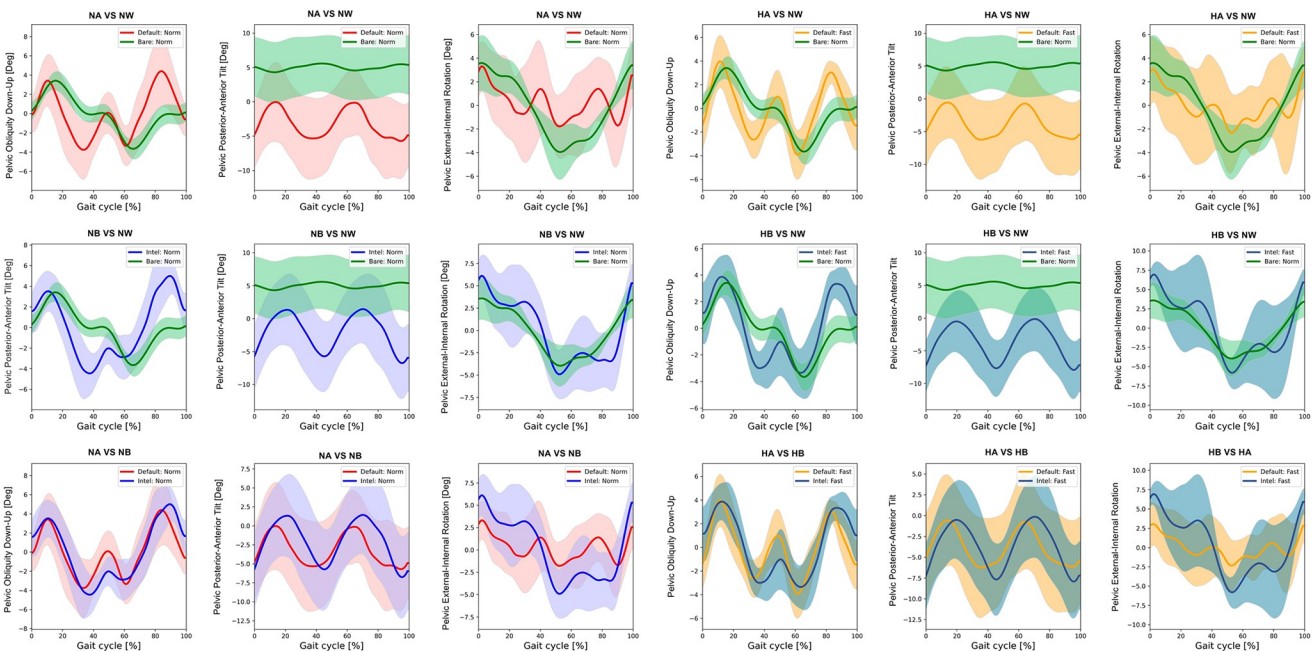

**Fig 2. Each graph shows the mean pelvic kinematic angle of exoskeleton mode versus normal walking and intelligent mode versus default mode of exoskeleton at normal and high speed.** (NA = normal-speed walking with the default-mode exoskeleton, NW = normal walking, NB = normal-speed walking with the intelligent-mode exoskeleton, HA = high-speed walking with the default-mode exoskeleton, HB = high-speed walking with the intelligent-mode exoskeleton).

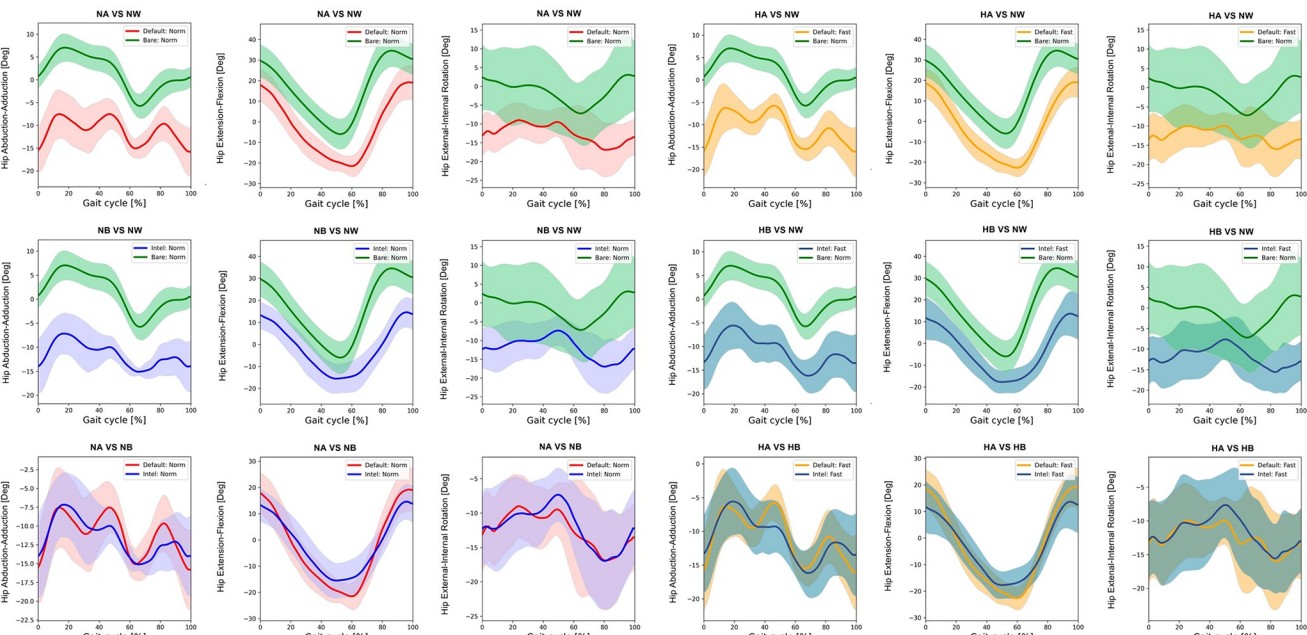

**Fig 3. Each graph shows the mean hip kinematic angle of exoskeleton mode versus normal walking and intelligent mode versus default mode of exoskeleton at normal and high speed.** (NA = normal-speed walking with the default-mode exoskeleton, NW = normal walking, NB = normal-speed walking with the intelligent-mode exoskeleton, HA = high-speed walking with the default-mode exoskeleton, HB = high-speed walking with the intelligent-mode exoskeleton).

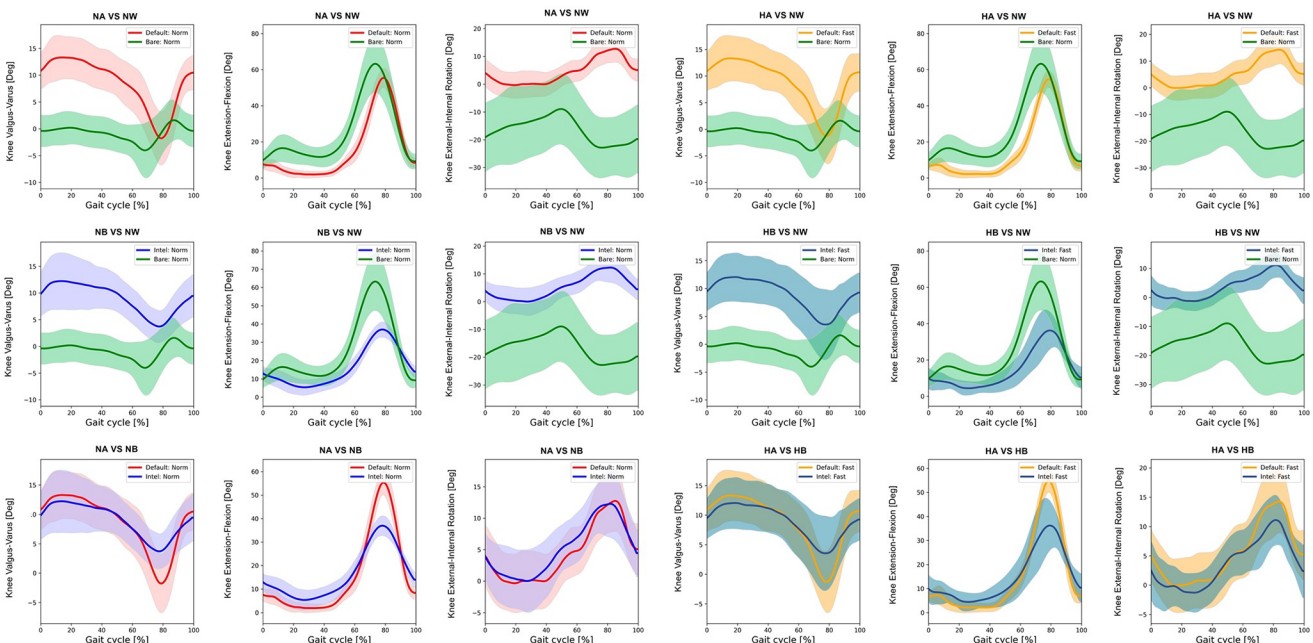

**Fig 4. Each graph shows the mean knee kinematic angle of exoskeleton mode versus normal walking and intelligent mode versus default mode of exoskeleton at normal and high speed.** (NA = normal-speed walking with the default-mode exoskeleton, NW = normal walking, NB = normal-speed walking with the intelligent-mode exoskeleton, HA = high-speed walking with the default-mode exoskeleton, HB = high-speed walking with the intelligent-mode exoskeleton).

advanced and unique for a wearer. Besides, compared to an exoskeleton from Cyberdyne using muscle signal as a trigger, its main purpose aims to create a gait movement with a body-supported device rather than solely assist a wearer energetically as for our Exo-H3 exoskeleton. Regarding to this study, walking with either the default or intelligent-mode exoskeleton

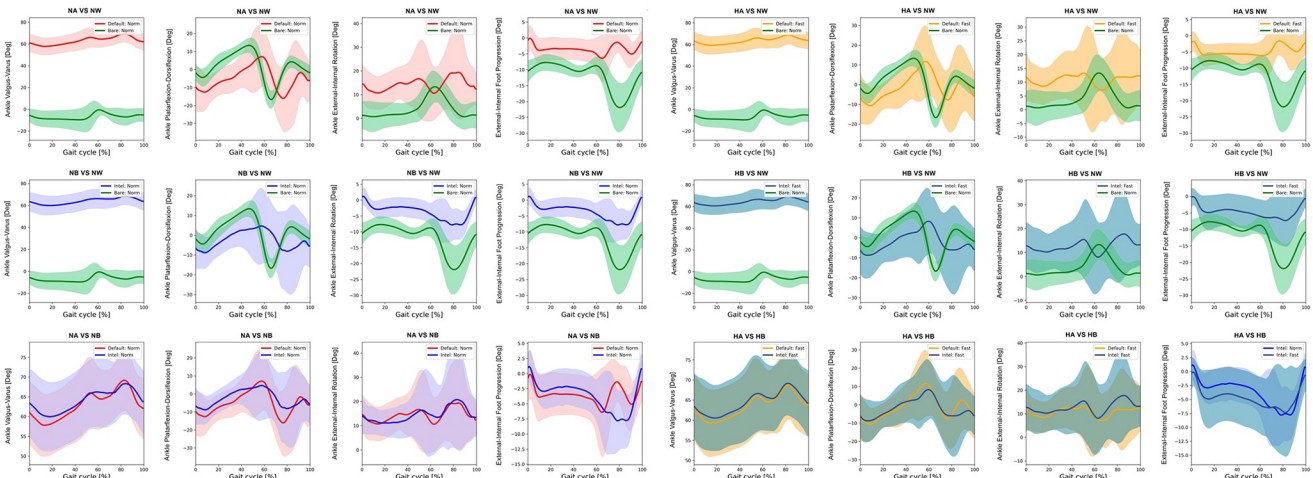

**Fig 5. Each graph shows the mean ankle kinematic angle of exoskeleton mode versus normal walking and intelligent mode versus default mode of exoskeleton at normal and high speed.** (NA = normal-speed walking with the default-mode exoskeleton, NW = normal walking, NB = normal-speed walking with the intelligent-mode exoskeleton, HA = high-speed walking with the default-mode exoskeleton, HB = high-speed walking with the intelligent-mode exoskeleton).

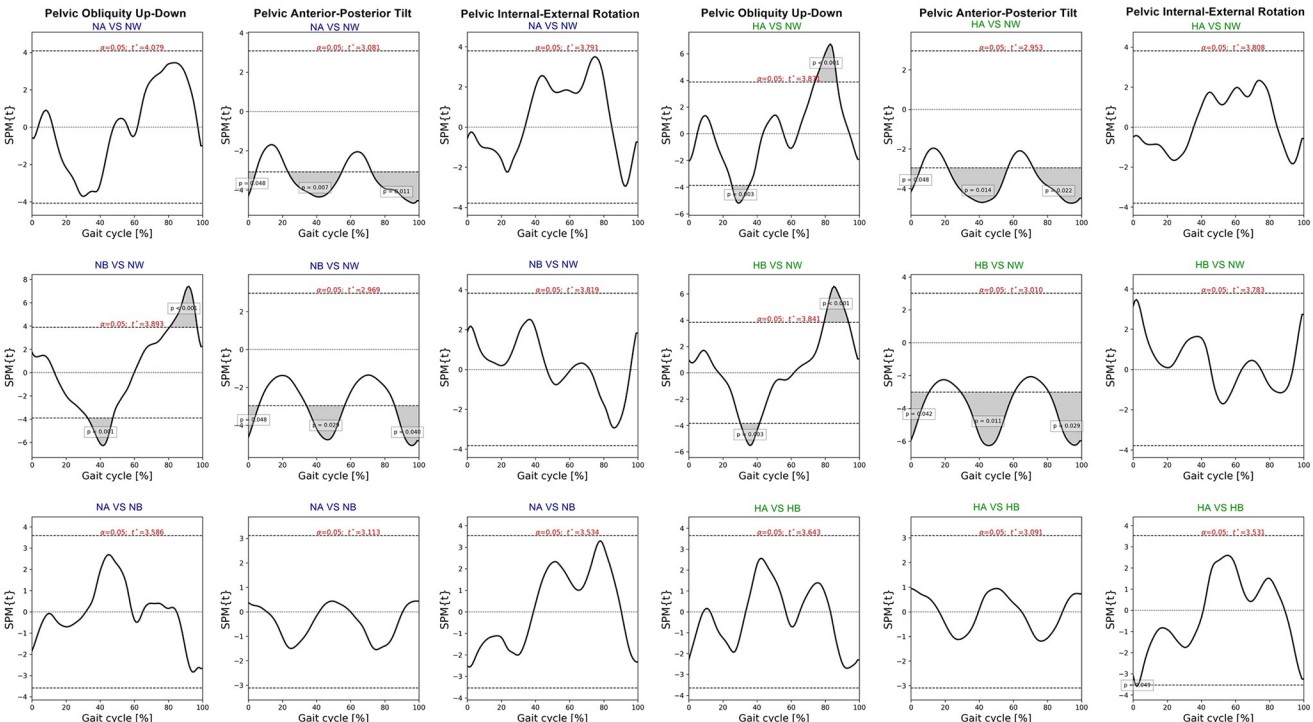

**Fig 6. Each graph shows SPM pair t-test analysis for the pelvic angle of exoskeleton mode versus normal walking and intelligent mode versus default mode of exoskeleton at normal and high speed.** Shaded grey areas indicate significant differences (p < 0.05), with corresponding p-values. (NA = normal-speed walking with the default-mode exoskeleton, NW = normal walking, NB = normal-speed walking with the intelligent-mode exoskeleton, HA = high-speed walking with the default-mode exoskeleton, HB = high-speed walking with the intelligent-mode exoskeleton).

significantly decelerated velocity and cadence when compared to normal walking at both speed (p<0.001). Moreover, the default setting compromised the percentage of left swing but increased step width in contrast to walking without the exoskeleton. Average baseline gait parameters demonstrated velocity 100 cm/s, stride length 1.2 m, and stance phase 61%. The applying exoskeleton resulted in 58–75 cm/s velocity reduction, 0.3–29 cm shorter stride length, and 4–10% longer stance phase. Despite temporospatial parameter variation attributed to different mechanical designed exoskeletons [33–36], age and neuromuscular involvement of participants [37, 38], the amounts of gait parameter changes in this study were comparable with the other multi-joint adaptive technology. In work by Swank, et al. [39], the robotic exoskeleton (EKSO) had a slower walking speed, shorter stride length, and longer double-limb support time than those without an exoskeleton. Comparing with and without Ekso in 15 health adults, average gait parameters showed velocity 31.0 ± 0.4 vs 132.0 ± 1.6 cm/s, stride length 0.72 ± 0.14 vs 1.41 ± 0.12 m, and double support 4.5 ± 0.6 vs 1.7 ± 0.2 s [39]. The explanations are the conflict between intrinsic and forced walking styles occurred when the subjects are forced to walk with fixed speed in the default mode. Our control algorithm plays an important role to mitigate this conflict by allowing the change to be occurred according to the subjects' intention when the exoskeleton forces them out of their comfortable walking style (stride length and velocity in this case). However, the exoskeleton-subject adaptation seems to be difficult in some cases comparing to the default mode at various speeds. The intelligence mode diminished the velocity approximately 9 cm/s and stride length by 22–23 cm when compared to the default exoskeleton at both speeds. The benefits of this algorithm for the real world are

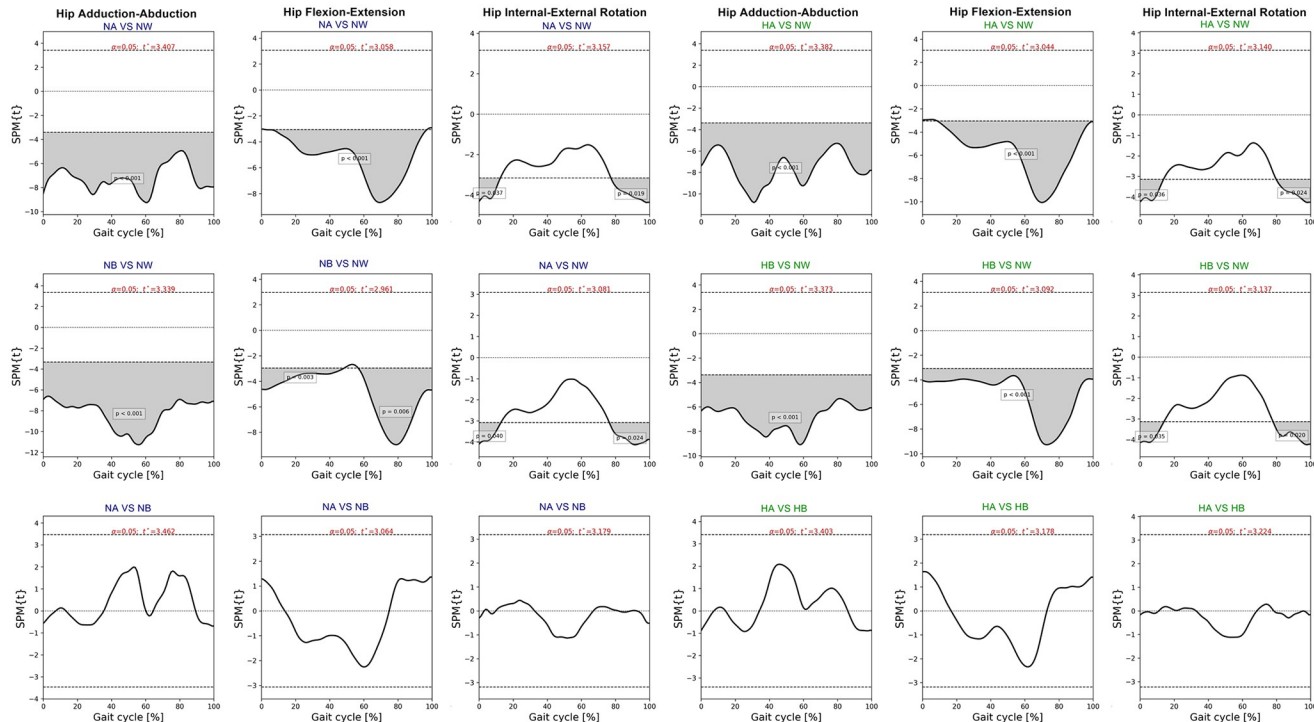

**Fig 7. Each graph shows SPM pair t-test analysis for the hip angle of exoskeleton mode versus normal walking and intelligent mode versus default mode of exoskeleton at normal and high speed.** Shaded grey areas indicate significant differences ($p < 0.05$), with corresponding p-values. (NA = normal-speed walking with the default-mode exoskeleton, NW = normal walking, NB = normal-speed walking with the intelligent-mode exoskeleton, HA = high-speed walking with the default-mode exoskeleton, HB = high-speed walking with the intelligent-mode exoskeleton).

probably more stability by controlling center of gravity [35], and facilitates comfortability and performance, especially walking at high speed (Table 4). The drawback of the intelligence setting is slowly forward moving.

Subjects walking with the exoskeleton increased posterior tilt of the pelvis, abduction-extension-external rotation of the hip, and varus-extension-internal rotation of the knee, as well as varus, internal rotation, and plantar flexion of the ankle. However, the intelligent mode produced more anterior pelvic tilt during midstance and mid-swing at both speed, hip abduction by -2.25 ±1.53 degrees during mid-swing at normal speed and decreased hip abduction by 2.24 ±2.09 degrees during heel strike at high speed, compared to the default. The increase in hip abduction may be necessary for lateral trunk movements to initiate the swing phase of limb advancement while walking with the exoskeleton [40]. Hip motions of the intelligent mode reduced flexion during heel strike and mid-swing and extension during midstance and terminal stance, compared to default mode and normal walking. Limited hip flexion during mid-swing and extension during terminal stance lessens propulsive force generation during the stance phase and impedes limb advancement during the swing phase [41]. This finding may arise from undesirable compensatory movements of intelligent-mode walking by hip hiking and circumduction.

Both exoskeletal modalities had different sagittal plane knee kinematics. Additionally, knee flexion during swing phase was lower in the intelligent mode than normal walking and the default. Similar to the previous study [39], wearing the exoskeleton diminished hip and ankle range of motion but facilitated knee movement during the stance phase comparing to walking

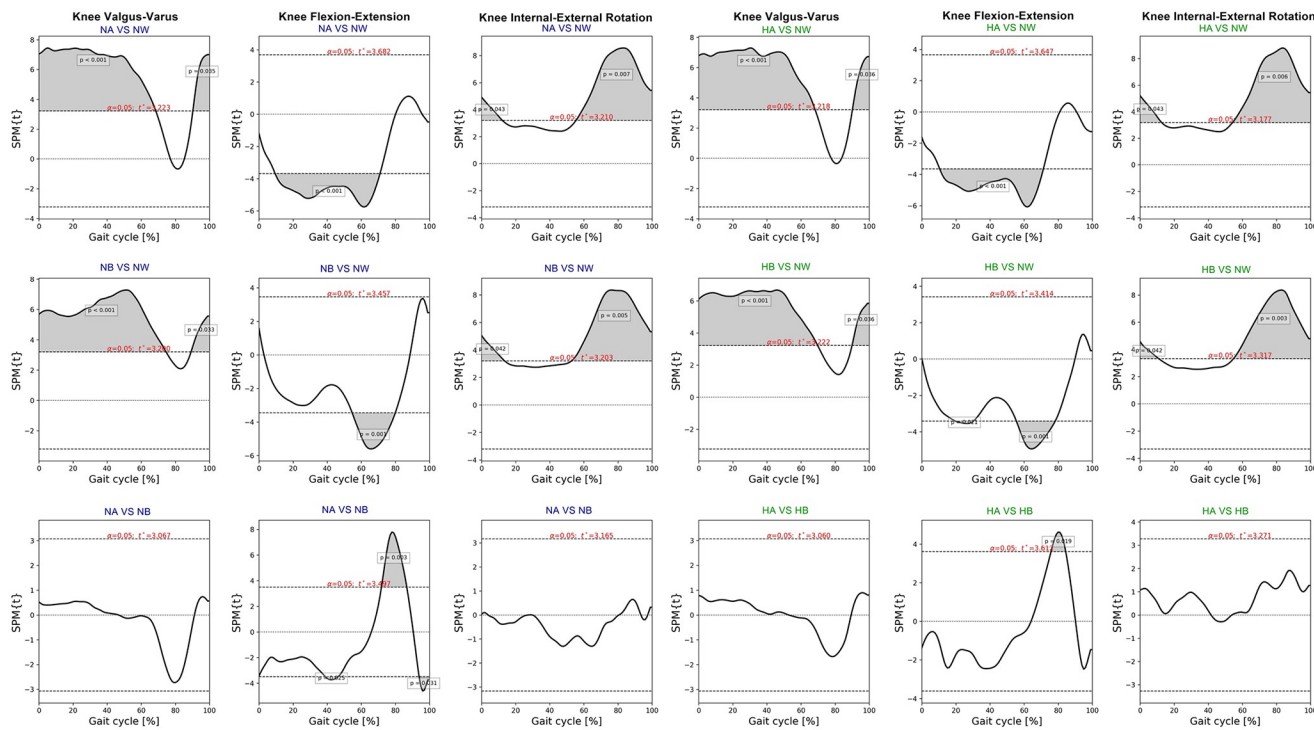

**Fig 8. Each graph shows SPM pair t-test analysis for the knee angle of exoskeleton mode versus normal walking and intelligent mode versus default mode of exoskeleton at normal and high speed.** Shaded grey areas indicate significant differences (p < 0.05), with corresponding p-values. (NA = normal-speed walking with the default-mode exoskeleton, NW = normal walking, NB = normal-speed walking with the intelligent-mode exoskeleton, HA = high-speed walking with the default-mode exoskeleton, HB = high-speed walking with the intelligent-mode exoskeleton).

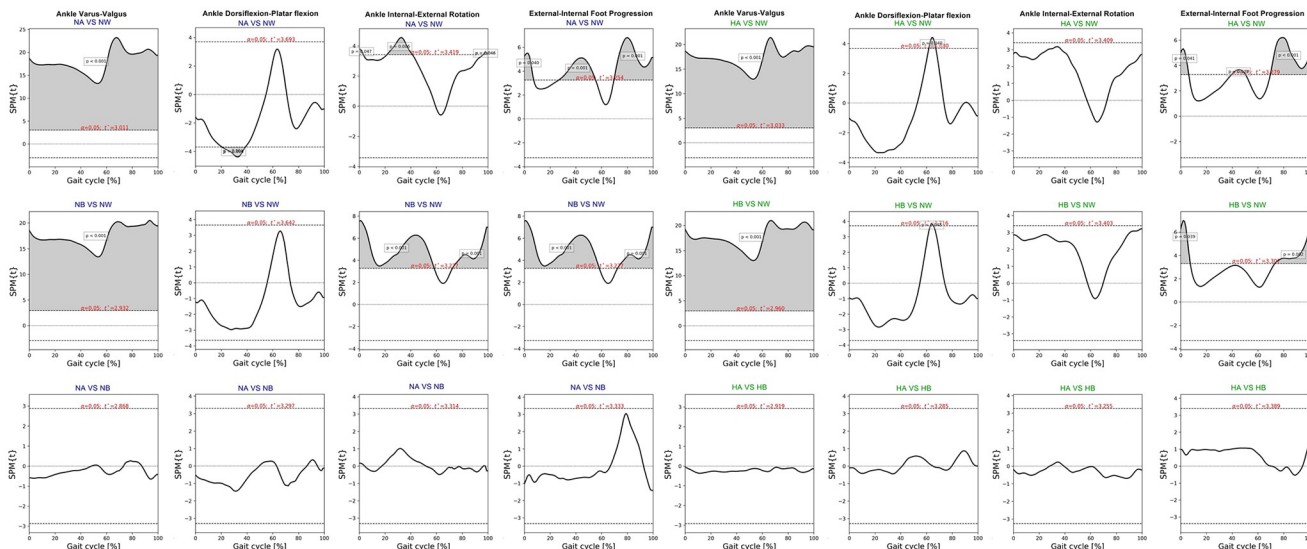

**Fig 9. Each graph shows SPM pair t-test analysis for the ankle angle of exoskeleton mode versus normal walking and intelligent mode versus default mode of exoskeleton at normal and high speed.** Shaded grey areas indicate significant differences (p < 0.05), with corresponding p-values. (NA = normal-speed walking with the default-mode exoskeleton, NW = normal walking, NB = normal-speed walking with the intelligent-mode exoskeleton, HA = high-speed walking with the default-mode exoskeleton, HB = high-speed walking with the intelligent-mode exoskeleton).

**Table 4. The comparison of muscle activity, VO₂max, user's comfort and exoskeleton performance score between the exoskeleton modes and normal walking.**

| | Normal speed | | | | | | High speed | | | | | |
|---|---|---|---|---|---|---|---|---|---|---|---|---|
| | NA vs NW | | NB vs NW | | NB vs NA | | HA vs NW | | HB vs NW | | HB vs HA | |
| | Mean difference (SD) | P-value | Mean difference (SD) | P-value | Mean difference (SD) | P-value | Mean difference (SD) | P-value | Mean difference (SD) | P-value | Mean difference (SD) | P-value |
| **Muscle activity (V) (%MVC)** | | | | | | | | | | | | |
| Right GMAX | -5.36 (26.13) | 0.575[a] | -7.58 (23.45) | 0.779[a] | -2.23 (10.29) | 0.779[a] | -5.61 (22.05) | 0.889[a] | -7.43 (26.58) | 0.833[a] | -1.82 (7.15) | 0.528[a] |
| Left GMAX | -12.89 (41.66) | 0.674[a] | -16.67 (41.42) | 0.575[a] | -3.78 (10.71) | 0.351 | -9.36 (32.63) | 0.779[a] | -15.42 (41.58) | 0.484[a] | -6.06 (10.29) | 0.233[a] |
| Right TA | 9.27 (18.10) | 0.191 | 10.79 (20.44) | 0.069 | 1.53 (10.91) | 0.704 | 5.59 (8.71) | 0.113 | -0.33 (12.54) | 0.943 | -5.92 (11.54) | 0.190 |
| Left TA | 8.34 (10.17) | 0.053 | 6.77 (10.55) | 0.113 | -1.58 (9.29) | 0.646 | 4.14 (6.87) | 0.132 | 1.12 (8.31) | 0.715 | -3.02 (4.43) | 0.095 |
| Right GA | 27.49 (45.10) | 0.128 | 44.35 (104.23) | 0.208[a] | 16.86 (114.28) | 0.263[a] | 13.62 (41.36) | 0.383 | 6.41 (45.14) | 0.700 | -7.21 (10.56) | 0.095 |
| Left GA | 122.23 (190.84) | 0.036 *[a] | 38.29 (63.13) | 0.161[a] | -83.94 (134.19) | 0.069[a] | 3.41 (14.35) | 0.524 | 7.73 (29.79) | 0.487 | 4.32 (27.32) | 0.668 |
| Right RF | 2.93 (22.43) | 0.779[a] | 4.06 (17.55) | 0.484[a] | 1.14 (12.43) | 0.802 | -1.98 (16.71) | 0.889[a] | -0.19 (16.05) | 0.779[a] | 1.78 (8.76) | 0.582 |
| Left RF | 7.90 (10.13) | 0.063 | 7.03 (14.00) | 0.199 | -0.88 (18.49) | 0.897 | 3.22 (10.83) | 0.428 | 1.97 (9.48) | 0.575 | -1.25 (7.36) | 0.646 |
| Right BF | 6.22 (15.36) | 0.289 | 4.61 (14.25) | 0.391 | -1.61 (5.24) | 0.413 | 5.14 (11.17) | 0.234 | 0.14 (11.36) | 0.974 | -5.00 (9.55) | 0.182 |
| Left BF | 11.66 (19.71) | 0.138 | 9.83 (19.03) | 0.188 | -1.83 (8.99) | 0.583 | 21.42 (43.16) | 0.036 *[a] | 17.20 (45.12) | 0.484[a] | -4.23 (8.59) | 0.292[a] |
| **VO₂ max (ml/kg/min)** | | | | | | | | | | | | |
| | 3.25 (5.46) | 0.136 | 1.45 (4.65) | 0.407 | -1.8 (4.25) | 0.270 | 2.68 (2.09) | 0.008* | 0.65 (2.87) | 0.543 | -2.03 (2.07) | 0.028* |
| **User comfort score** | | | | | | | | | | | | |
| | | | | | -0.08 (0.35) | 0.517 | | | | | 0.06 (0.18) | 0.351 |
| **Exoskeleton performance score** | | | | | | | | | | | | |
| | | | | | -0.02 (0.74) | 0.954 | | | | | 0.09 (0.26) | 0.336 |

NA = normal-speed walking with the default-mode exoskeleton, NB = normal-speed walking with the intelligent-mode exoskeleton, HA = high-speed walking with the default-mode exoskeleton, HB = high-speed walking with the intelligent-mode exoskeleton, SD = standard deviation, GMAX = gluteus maximus, TA = tibialis anterior, GA = gastrocnemius, RF = rectus femoris, BF = biceps femoris,

*P-value <0.05, P-value from Paired t-test,

[a]P-value from Wilcoxon signed-rank test.

without using a robotic exoskeleton. The compensation of the intelligent exoskeleton mode through increased hip abduction may result from limiting knee flexion. Meanwhile, walking with higher knee flexion in the default mode may be a compensatory mechanism for restricted ankle dorsiflexion.

Regarding muscle activity, the default exoskeleton raised left gastrocnemius firing at normal speed (122.23 ± 190.84V) and left biceps femoris at high speed (21.42 ± 43.16V) compared to walking without the exoskeleton (p = 0.036). On the other hand, the intelligent mode had comparable muscle activity with normal walking, and significantly lower oxygen consumption by -2.03 ± 2.07 ml/kg/min at high speed than the default. These findings indicate advantages of

the intelligent mode at high speed, i.e., less effort of walking by reducing muscle activity and oxygen consumption. Minimizing the metabolic consumption of the wearer may result from the compatibility of kinematics [42]. The intelligent mode at high-speed walk tended to be superior to the default in terms of comfortable use and performance of the exoskeleton. Nevertheless, these scores did not reach statistical significance between exoskeleton groups.

### 4.1 Limitations

This study has several limitations according to the exoskeleton model, motion analysis marker placement, questionnaires, and generalizability. In regards with the exoskeleton's design, the joint motors obscured the bony landmarks such as the S1 spine, the lateral aspect of the knees, and lateral malleoli. Reflective markers were attached to the exoskeleton closed to specific locations as possible to avoid inaccurate kinematic analysis. The comfort and performance scores were modified from the previous studies, and results may apply with cautions because of this unique exoskeletal setting, including only healthy adults with forearm crutches assistance, and very small number of sample size contributed to major limitation.

## 5. Conclusion

Adaptive exoskeleton as an intelligent mode improves muscle activity, oxygen consumption, user comfort and performance of the device. Further large studies are needed to refine the temporal-spatial, lower extremity kinematics, particularly hip-knee-ankle sagittal motions, to synchronize body and exoskeleton mobility. Besides interaction torque reduction by joint angle control adaptation, future research is suggested for both hardware and software. For the former, replacing the full-body exoskeleton to modular type e.g., knee exoskeleton may facilitate more flexible motion, and various hip-knee-ankle module combination. For the latter, free-mode or zero-torque control is required in order to learn more human-like patterns (kinematics). Also, applying CPG and radius basis function (RBF) to learn different frequencies and generate different shape/pattern will allow more changes in frequency and joint angle pattern.

## Supporting information

**S1 Checklist. CONSORT checklist.**
(DOC)

**S1 Text. Study protocol.**
(PDF)

**S1 File. The detail of Exoskeleton and the development of intelligent exoskeleton.**
(DOCX)

**S2 File. The speed of Exo-H3.**
(DOCX)

**S3 File. Reflective markers based on modified Helen Hayes.**
(DOCX)

**S1 Table. Characteristic of the participants.**
(DOCX)

**S2 Table. The comparison of lower limb kinematics between groups during walking without the exoskeleton.**
(DOCX)

**S3 Table. The comparison of muscle activity (%RMS) between groups during walking without the exoskeleton.**
(DOCX)

**S1 Data.**
(XLSX)

## Author Contributions

**Conceptualization:** Krongkaew Supapitanon, Tanyaporn Patathong, Chaicharn Akkawutvanich, Poramate Manoonpong, Patarawan Woratanarat, Chanika Angsanuntsukh.

**Data curation:** Krongkaew Supapitanon, Tanyaporn Patathong, Chaicharn Akkawutvanich, Arthicha Srisuchinnawong, Worachit Ketrungsri, Poramate Manoonpong, Patarawan Woratanarat, Chanika Angsanuntsukh.

**Formal analysis:** Krongkaew Supapitanon, Tanyaporn Patathong, Chaicharn Akkawutvanich, Poramate Manoonpong, Patarawan Woratanarat, Chanika Angsanuntsukh.

**Funding acquisition:** Poramate Manoonpong.

**Investigation:** Krongkaew Supapitanon, Tanyaporn Patathong, Chaicharn Akkawutvanich, Poramate Manoonpong, Patarawan Woratanarat, Chanika Angsanuntsukh.

**Methodology:** Krongkaew Supapitanon, Tanyaporn Patathong, Chaicharn Akkawutvanich, Arthicha Srisuchinnawong, Poramate Manoonpong, Patarawan Woratanarat, Chanika Angsanuntsukh.

**Supervision:** Poramate Manoonpong, Patarawan Woratanarat, Chanika Angsanuntsukh.

**Validation:** Krongkaew Supapitanon, Tanyaporn Patathong, Chaicharn Akkawutvanich, Poramate Manoonpong, Patarawan Woratanarat, Chanika Angsanuntsukh.

**Writing – original draft:** Krongkaew Supapitanon, Tanyaporn Patathong, Chaicharn Akkawutvanich, Poramate Manoonpong, Patarawan Woratanarat, Chanika Angsanuntsukh.

**Writing – review & editing:** Krongkaew Supapitanon, Tanyaporn Patathong, Chaicharn Akkawutvanich, Poramate Manoonpong, Patarawan Woratanarat, Chanika Angsanuntsukh.

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
