## [Decision Letter · Decision Letter 0]

19 Aug 2024

PONE-D-24-25513Comprehensive Multi-Metric Analysis of User Experience and Performance in Adaptive and Non-adaptive Lower-limb ExoskeletonsPLOS ONE

Dear Dr. Angsanuntsukh,

Thank you for submitting your manuscript to PLOS ONE. After careful consideration, we feel that it has merit but does not fully meet PLOS ONE’s publication criteria as it currently stands. Therefore, we invite you to submit a revised version of the manuscript that addresses the points raised during the review process.

The manuscript evaluates the performance and user experience of adaptive versus non-adaptive lower-limb exoskeletons using various metrics. While the study is well-conceived, reviewers raised significant concerns, particularly regarding power calculations, missing data handling, and clarity in reporting. Additional issues include a need for a more comprehensive introduction, clearer articulation of design parameter impacts, and consideration of alternative control methods. Reviewers also suggested improving the readability of results and ensuring the manuscript is thoroughly polished for errors.

We look forward to receiving your revised manuscript.

Kind regards,

Jyotindra Narayan

Academic Editor

PLOS ONE

Journal Requirements:

3. Thank you for stating the following financial disclosure: This work was supported by the Vidyasirimedhi Institute of Science & Technology (VISTEC) under the EXOVIS project (Grant No. I20POM-INT010). 

4. Thank you for stating the following in the Acknowledgments Section of your manuscript: This work was supported by the Vidyasirimedhi Institute of Science & Technology (VISTEC) under the EXOVIS project (Grant No. I20POM-INT010).

Please remove any funding-related text from the manuscript and let us know how you would like to update your Funding Statement. Currently, your Funding Statement reads as follows: This work was supported by the Vidyasirimedhi Institute of Science & Technology (VISTEC) under the EXOVIS project (Grant No. I20POM-INT010).

Reviewers' comments:

Reviewer's Responses to Questions

**Comments to the Author**

1. Is the manuscript technically sound, and do the data support the conclusions?

Reviewer #1: Yes

Reviewer #2: No

Reviewer #3: Partly

2. Has the statistical analysis been performed appropriately and rigorously? 

Reviewer #1: Yes

Reviewer #2: Yes

Reviewer #3: Yes

3. Have the authors made all data underlying the findings in their manuscript fully available?

Reviewer #1: Yes

Reviewer #2: Yes

Reviewer #3: Yes

4. Is the manuscript presented in an intelligible fashion and written in standard English?

Reviewer #1: Yes

Reviewer #2: Yes

Reviewer #3: Yes

5. Review Comments to the Author

Reviewer #1: The manuscript entitled “Comprehensive Multi-Metric Analysis of User Experience and Performance in Adaptive and Non-adaptive Lower-limbExoskeletons” evaluates the user experience and performance of adaptive versus non-adaptive lower-limb exoskeletons through a multi-metric analysis involving gait, muscle activity, oxygen consumption, and comfort. Using a crossover design with healthy adults, findings revealed that adaptive exoskeletons, utilizing central pattern generator-based controls, exhibited lower velocity and stride lengths but offered advantages in specific kinematic adjustments and reduced oxygen consumption, suggesting a need for further refinement in exoskeleton design to enhance user comfort and performance.The study is well-developed and the results are intriguing. However, the attached concerns need to be addressed.

Reviewer #2: As the statistical reviewer I will focus on methods and reporting.

Major

1) I struggled to follow the power calculations, or they are wrong. much more clarity is needed. if the means and SDs are reported and the authors calculated mean differences, i arrive at a very different sample calculation. Also it was not clear if an appropriate crossover design power calculation approach was used. This is highly problematic. also, even if this was correct for the primary outcome, it is extremely unlikely that power is adequate for other outcomes which is a major limitation.

2) how did the authors deal with missing data, can they expand to clarify in the methods section as per the CONSORT statement?

Minor

1) Can this sentence be rephrased for clarity and grammer please: "The registration was delayed due to a misunderstanding between the project and the principal investigator resigned from the organization."

2) Stata not STATA (not an acronym).

Reviewer #3: This paper compares between adaptative and non-adaptative control laws for robotic exoskeletons used in rehabilitation of lower limbs, using different metrics. Authors pointed out the interest in the CPG-based controllers, its robustness to perturbations, compared to other control methods.

By and large, the idea presented in this paper is attractive. However, several comments are raised and should be well addressed to improve this paper:

1- The introduction is very short (compared to the abstract) and should be clearly expanded. Moreover, in introduction, it is not enough to state the current work. It should be expended and reconstructed. Including the motivation, the main difficulties, the main work and the improvements compared with previous related works should be emphasized in this section.

2- The authors should polish the paper suitably. The whole paper should be reviewed carefully, in order to correct all the typing errors.

3- The authors should give a Remark to illustrate how the design parameters effect the control performance of the exoskeleton in low and high speeds, and how to choose these parameters.

4- As a clinical trial study, how you quantity the novelty of the proposed method ?

5- Related to previous question, there are some control methods that are based on SMC and also those based on linear matrix inequality to design gains and improve performances according to some gains. These SMC- and LMI-based two control methods have been considered recently for lower limber exoskeletons. I suggest to see for example the following papers “An LMI-based robust state-feedback controller design for the position control of a knee rehabilitation exoskeleton robot: Comparative analysis”, “Fast terminal sliding mode control with rapid reaching law for a pediatric gait exoskeleton system”, “Adaptive backstepping sliding mode subject-cooperative control for a pediatric lower-limb exoskeleton robot”, “Design and Validation of a Pediatric Gait Assistance Exoskeleton System with Fast Non-Singular Terminal Sliding Mode Controller”

6- There are several figures included in the results. It is good, but it is difficult to read them easily. Then, I suggest to add Tables to recapitulate the necessary and important details from these figures

6. PLOS authors have the option to publish the peer review history of their article (what does this mean?). If published, this will include your full peer review and any attached files.

Reviewer #1: No

Reviewer #2: No

Reviewer #3: No

---

## [Author Response · Author response to Decision Letter 0]

18 Oct 2024

Response to reviewers’ comments 

Reviewer #1: The manuscript entitled “Comprehensive Multi-Metric Analysis of User Experience and Performance in Adaptive and Non-adaptive Lower-limb Exoskeletons” evaluates the user experience and performance of adaptive versus non-adaptive lower-limb exoskeletons through a multi-metric analysis involving gait, muscle activity, oxygen consumption, and comfort. Using a crossover design with healthy adults, findings revealed that adaptive exoskeletons, utilizing central pattern generator-based controls, exhibited lower velocity and stride lengths but offered advantages in specific kinematic adjustments and reduced oxygen consumption, suggesting a need for further refinement in exoskeleton design to enhance user comfort and performance. The study is well-developed and the results are intriguing. However, the attached concerns need to be addressed. 

1. Methodological Rigor: The study design is a crossover trial; however, can you clarify if there were any washout periods between conditions to prevent carry-over effects from one exoskeleton mode to the other? Additionally, how was the sample size determined, and could you provide more details on the statistical power analysis?

Answer for wash out periods: 

 Thank you for your questions. Regarding the washout period, we ensured that each participant had a minimum of 10 minutes of rest as a washout period before moving on to the next exoskeleton mode. The transition to the next mode only occurred when the participant felt comfortable and ready to proceed. This approach aligns with the study of Hybart et al. (2023), where the authors stated, "Between each condition, the participants rest as needed." Based on this reference, we believe the washout period we implemented is sufficient to prevent carry-over effects between conditions.

Revision:

 I have revised the content in the manuscript line 164-165

 From

 “A 10-minute resting period was implemented between each intervention to reduce the carryover effect.”

 To

 “The participants had a minimum of 10 minutes of rest [23], or more if needed, between each intervention to minimize carryover effects [24].” 

Answer for sample size calculation:

 For sample size calculation, we followed the guidelines provided by Assoc. Prof. Cameron Hurst from Chulalongkorn University. We used the sample size calculation tool available at this website https://www2.ccrb.cuhk.edu.hk/stat/mean/tsmc_sup.htm After filling the required parameters specifically, a mean difference of 6.8 degrees from the study of Guzik et al.(2020), a standard deviation (SD) of 7.6, and a margin of 5 degrees—the calculated total sample size was 8 participants for the crossover design. 

Revision:

 I have revised the content of sample size calculation in the manuscript line 227-232

 From

 “Sample size estimation was performed based on evidence from the mean and standard deviations of peak value of knee extension of the unaffected leg in stroke patients with Knee Ankle Foot Orthosis (KAFO) (-0.8±7.6 degrees) compared to those without KAFO (0.2±7.2 degrees). The study used a type I error (alpha) of 0.05 and power of 0.8. Therefore, a sample size was 4 participants per group.”

 To

 “Sample size estimation was conducted using a sample size calculator for crossover design (https://www2.ccrb.cuhk.edu.hk/ stat/mean/tsmc_sup.htm). The estimation was based on a mean difference of 6.8 degrees [29], a standard deviation (SD) of 7.6 degrees, and a margin of 5 degrees. The study used a type I error (alpha) of 0.05 and power of 0.8. Therefore, the total sample size required for the study was 8 participants.” 

Reference:

 R. L. Hybart and D. P. Ferris, "Neuromechanical Adaptation to Walking With Electromechanical Ankle Exoskeletons Under Proportional Myoelectric Control," in IEEE Open Journal of Engineering in Medicine and Biology, vol. 4, pp. 119-128, 2023, doi: 10.1109/OJEMB.2023.3288469

 Guzik A, Drużbicki M, Wolan-Nieroda A, Turolla A, Kiper P. Estimating Minimal Clinically Important Differences for Knee Range of Motion after Stroke. J Clin Med. 2020 Oct 15;9(10):3305. doi: 10.3390/jcm9103305. PMID: 33076214; PMCID: PMC7602397.

2. Control Conditions and Variables: In comparing adaptive and non-adaptive exoskeletons, how did you control for external variables such as participants' physical condition and previous experience with similar devices? How might these factors have influenced the outcomes? 

Answer: 

 The research team used strict inclusion criteria to select participants who were healthy, had no walking impairments, and had no prior experience using similar devices. On the day of the experiment, each participant was trained under expert supervision to walk with the exoskeleton until they were confident and able to walk independently. Additionally, this study employed a crossover design, which compared outcomes within the same subjects, thereby ensuring that prognostic factors were comparable and minimizing the influence of external variables.

3. Results Interpretation: You reported significant differences in stride length and velocity

between the two modes. How do these differences translate into practical benefits or

drawbacks for users in real-world scenarios?

Answer: 

 From our data, the intelligence mode diminished the velocity approximately 9 cm/s and stride length by 22-23 cm when compared to the default exoskeleton at both speeds. The benefits for the real world are probably more stability by controlling center of gravity (Qiu S, 2023), and facilitates comfortability and performance, especially at high speed. The drawback of the intelligence setting is slowly forward moving. 

The explanations are the conflict between intrinsic and forced walking styles occurred when the subjects are forced to walk with fixed speed in the default mode. Our control algorithm can adapt to different users which can be seen from the results showing different stride lengths and velocities. Normally, each subject has his/her own intrinsic preferred swing frequency (which can be indirectly translated into the stride length) and the walking speed (velocity) when the subjects walk freely without wearing the exoskeleton. When the subjects are forced to walk in the default mode where the speed is fixed and cannot be altered during walking, the conflict between intrinsic and forced walking styles might occur. Our control algorithm plays an important role to mitigate this conflict by allowing the change to be occurred according to the subjects’ intention when the exoskeleton forces them out of their comfortable walking style (stride length and velocity in this case). For example, one subject may walk naturally fast, but he/she feels insecure while assisting by the exoskeleton at the high speed and preferred to tone it down which is allowed by the algorithm. However, when both things (the exoskeleton and the subject) adapt simultaneously for each other, this seems to be difficult in some cases comparing to human adaptation only (in case of default mode, various speeds).

References

1. Qiu, S., Pei, Z., Wang, C. et al. Systematic Review on Wearable Lower Extremity Robotic Exoskeletons for Assisted Locomotion. J Bionic Eng 20, 436–469 (2023). https://doi.org/10.1007/s42235-022-00289-8

Revision: Revision was made in the Discussion part

From 

 “Regarding to this study, walking with either the default or intelligent-mode exoskeleton significantly decelerated velocity and cadence when compared to normal walking at both speed (p<0.001). Moreover, the default setting compromised the percentage of left swing but increased step width in contrast to walking without the exoskeleton.” 

To

 “Line 417-421: Regarding to this study, walking with either the default or intelligent-mode exoskeleton significantly decelerated velocity and cadence when compared to normal walking at both speed (p<0.001). Moreover, the default setting compromised the percentage of left swing but increased step width in contrast to walking without the exoskeleton”. 

 “Line 431-440: The explanations are the conflict between intrinsic and forced walking styles occurred when the subjects are forced to walk with fixed speed in the default mode. Our control algorithm plays an important role to mitigate this conflict by allowing the change to be occurred according to the subjects’ intention when the exoskeleton forces them out of their comfortable walking style (stride length and velocity in this case). However, the exoskeleton-subject adaptation seems to be difficult in some cases comparing to the default mode at various speeds.The intelligence mode diminished the velocity approximately 9 cm/s and stride length by 22-23 cm when compared to the default exoskeleton at both speeds. The benefits of this algorithm for the real world are probably more stability by controlling center of gravity [24], and facilitates comfortability and performance, especially walking at high speed (Table 4). The drawback of the intelligence setting is slowly forward moving.”

4. Technical Specifications: Could you provide more detailed specifications of the adaptive control mechanisms in the exoskeletons? How do these mechanisms adapt in real-time to the user's gait dynamics, and what algorithms support this adaptability?

Answer: 

 Thank you so much for your suggestion. The detailed specification of adaptive control mechanisms and its effects towards user’s kinematics has already provided in the supplementary file (S1 File) as following

For the default control mode (Fig 2A), joint position patterns (θ ®) of hip, knee, and ankle for both legs are obtained from the company. These joint patterns are average profiles collected from European subjects. When assist, our mid-level control algorithm samples the joint profile and directly sends each joint angle data (θ) in an appropriate time to moderate a swing speed without any feedback information (open-loop control). The default control mechanism is designed for gait generation without adaptation during locomotion.

 For the intelligent control mode [3], it performs as closed-loop control based on six neural control modules (modules 1-6, Fig 2B.) with interconnected feedback. Each module is designed to interact with other modules uniquely. The hip and knee modules share their feedback pathways (δ) to all the other modules, while ankle modules only receive adaptation signals from the others. This coupling structure with bidirectional hip and knee feedback pathways is adopted to ensure a 90-degree phase difference between the left and right legs. The joint module (e.g., left hip module as an example in Fig 2B.) uses tracking loop concept where the algorithm adjusts its previous joint control signal to be suitable for a wearer in a current step. The module consists of six key components: phase generator, pattern generator, transformation equation (forward dynamics), time constant, gradient-based adaptation, and coupled cooperative primitives (CCP) adaptation. The phase generator creates rhythmic phase signals (C_1, C_2) based on a central pattern generator (CPG) concept. Those signals then activate the pattern generator module to produce a reference joint position angle at each time step (f) in this case for the left hip whose total profile/pattern is the joint position profile (θ ®) from the factory that has been previously learned and remembered by the module via weights of neural networks. The transformation eq. is then introduced with time constant block to create joint compliance on the system by making changes (τθ ˙) on top of the reference joint position angle (f). An adaptation part comes from changing tracking error (e= θ-θ^*) into two forms to be used by the gradient-based adaptation and the CCP adaptation modules. The former uses the error to adjust phase of the control signal, while the latter regulates the joint compliant path. Note that the phase gradient adaptation signal (δ) does not come only from the working module itself (δ_LH), but also from the other modules (δ_(other modules)) as mentioned earlier on the pathways. All in all, the intelligent control mechanism is designed for gait generation and adaptation in response to the tracking error during locomotion.

The algorithms support its adaptability was elaborated as Fig. 2 in the supplementary file (S1 File) as

Fig 2. (A) Block diagram of default control algorithm in this study. The factory joint position profile is used to drive the exoskeleton with adjustable swing speed. (B) Block diagram of intelligent control algorithm in this study. It is Adaptive Modular Neural Control (AMNC) for online gait synchronization. RH/LH = Right/Left hip, RK/LK = Right/Left knee, and RA/LA = Right/left ankle.

5. Biomechanical Measurements: What methods were used to measure kinematics and muscle activity, and how did you ensure the accuracy and reliability of these measurements, especially given the potential for interference from the exoskeleton itself?

Answer: 

 To measure kinematics and spatio-temporal parameters, we employed a highly reliable eight-camera motion analysis system from Motion Analysis Corporation (Santa Rosa, CA, USA), which is widely used in research for its accuracy and reliability. Muscle activity was assessed using surface electromyography (sEMG) with ProEMG software operating at 2000 Hz, alongside a Myon 320 wireless EMG system (Myon AG, Schwarzenberg, Switzerland), both of which are well-regarded in the field for their precision and have been utilized in numerous studies. To ensure the accuracy and reliability of our measurements, we performed calibration of these systems before each data collection session. This crucial step accounted for any potential interference from the exoskeleton and maintained the integrity of the collected data. We hope this clarifies the methods used and the steps taken to ensure the reliability of our measurements.

Revision:

 I have revised the manuscript by adding the sentence, 'To ensure the accuracy and reliability of our measurements, we performed calibration of these systems before each data collection session,' to lines 183-184.

6. User Experience: The study measures user comfort and performance scores; however, can you elaborate on how these were quantified? What specific aspects of user experience were evaluated, and how were these aspects chosen?

Answer: 

 Thank you so much for your queries. The user experience was evaluated using an exoskeletal performance questionnaire that comprised two domains: pain (while walking, rising from sitting, and standing) and difficulty (in rising from sitting, standing, and walking for distance). These aspects of user experience were selected based on the anticipated activities that users would engage in on the day of the study, such as rising from sitting, standing, and walking with the exoskeleton. The questions were chosen in advance to ensure their relevance to these activities, in accordance with our protocol focused on gait measurement. This approach allowed us to capture the most relevant experiences of discomfort (pain) and difficulty, providing a comprehensive assessment of the user experience with the exoskeleton.

The exoskeletal performance questionnaire comprised 2 domains; pain (walking, rising from sitting, standing) and difficulty (rising from sitting, standing, forward bending, walking, walking for distance). Each question was rated by respondents ranged from 1 (minimal issues) to 5 (significant problems). The total performance score was defined as best (8 points), good (9-23 points), moderate (24 points), poor (25-39 points), and worst (40 points). 

Also, the comfort assessment reference (J. F. Knight, C. Baber, A. Schwirtz and H. W. Bristow, "The comfort assessment of wearable computers," Proceedings. Sixth International Symposium on Wearable Computers, Seattle, WA, USA, 2002, pp. 65-72, doi: 10.1109/ISWC.2002.1167220) has been cited.

Revision: The revision was performed, line 213-226

From 

 “For the wearer's opinion to the device, their perceived change, emotion on self-image, anxiety on security, harm or painful, attachment, and movement were evaluated as the comfort score. Each item rated from a 1 (the least problem) to 5 (the most problem). Sum of total score was 30, and categorized as very comfortable (

---

## [Decision Letter · Decision Letter 1]

29 Oct 2024

Comprehensive Multi-Metric Analysis of User Experience and Performance in Adaptive and Non-adaptive Lower-limb Exoskeletons

PONE-D-24-25513R1

Dear Dr. Angsanuntsukh,

We’re pleased to inform you that your manuscript has been judged scientifically suitable for publication and will be formally accepted for publication once it meets all outstanding technical requirements.

Kind regards,

Jyotindra Narayan

Academic Editor

PLOS ONE

Additional Editor Comments (optional):

The reviewers have seen the revised work and recommended the work for publication. The manuscript can be accepted. Congratulations to the authors for keeping up the good work.

Reviewers' comments:

Reviewer's Responses to Questions

**Comments to the Author**

1. If the authors have adequately addressed your comments raised in a previous round of review and you feel that this manuscript is now acceptable for publication, you may indicate that here to bypass the “Comments to the Author” section, enter your conflict of interest statement in the “Confidential to Editor” section, and submit your "Accept" recommendation.

Reviewer #1: All comments have been addressed

Reviewer #2: All comments have been addressed

Reviewer #3: All comments have been addressed

2. Is the manuscript technically sound, and do the data support the conclusions?

Reviewer #1: Yes

Reviewer #2: Yes

Reviewer #3: Yes

3. Has the statistical analysis been performed appropriately and rigorously? 

Reviewer #1: Yes

Reviewer #2: Yes

Reviewer #3: Yes

4. Have the authors made all data underlying the findings in their manuscript fully available?

Reviewer #1: Yes

Reviewer #2: Yes

Reviewer #3: (No Response)

5. Is the manuscript presented in an intelligible fashion and written in standard English?

Reviewer #1: Yes

Reviewer #2: Yes

Reviewer #3: Yes

6. Review Comments to the Author

Reviewer #1: The paper has significantly improved, and the authors have addressed the concerns. The revised paper is highly recommended for publication.

Reviewer #2: I am satisfied with the authors' responses and the resulting changes to the paper...................

Reviewer #3: The paper has been clearly improved according to the comments of the reviewers. It can be then accepted.

7. PLOS authors have the option to publish the peer review history of their article (what does this mean?). If published, this will include your full peer review and any attached files.

Reviewer #1: No

Reviewer #2: No

Reviewer #3: No

---

## [Editor Report · Acceptance letter]

12 Nov 2024

PONE-D-24-25513R1 

PLOS ONE

Dear Dr. Angsanuntsukh, 

I'm pleased to inform you that your manuscript has been deemed suitable for publication in PLOS ONE. Congratulations! Your manuscript is now being handed over to our production team.

Kind regards, 

on behalf of

Dr. Jyotindra Narayan 

Academic Editor

PLOS ONE